# Plasmonic Nanomaterials in Dark Field Sensing Systems

**DOI:** 10.3390/nano13132027

**Published:** 2023-07-07

**Authors:** Wenjia Zhang, Xingyu Zi, Jinqiang Bi, Guohua Liu, Hongen Cheng, Kexin Bao, Liu Qin, Wei Wang

**Affiliations:** 1Tianjin Research Institute of Water Transport Engineering, M.O.T., Tianjin 300456, China; bijq@tiwte.ac.cn (J.B.); baokx@tiwte.ac.cn (K.B.); qinliu@tiwte.ac.cn (L.Q.); wangw@tiwte.ac.cn (W.W.); 2National Engineering Research Center of Port Hydraulic Construction Technology, Tianjin 300456, China; 3College of Microelectronics, Nankai University, Tianjin 300350, China; 1120220155@mail.nankai.edu.cn (X.Z.); 2120220335@mail.nankai.edu.cn (H.C.); 4Tianjin Key Laboratory of Optoelectronic Sensor and Sensing Network Technology, Tianjin 300350, China; 5School of Marine Science and Technology, Tianjin University, Tianjin 300192, China

**Keywords:** local surface plasmon resonance (LSPR), dark-field microscopy (DFM), plasmonic nanoparticles, quantitative detection

## Abstract

Plasma nanoparticles offer promise in data storage, biosensing, optical imaging, photoelectric integration, etc. This review highlights the local surface plasmon resonance (LSPR) excitation mechanism of plasmonic nanoprobes and its critical significance in the control of dark-field sensing, as well as three main sensing strategies based on plasmonic nanomaterial dielectric environment modification, electromagnetic coupling, and charge transfer. This review then describes the component materials of plasmonic nanoprobes based on gold, silver, and other noble metals, as well as their applications. According to this summary, researchers raised the LSPR performance of composite plasmonic nanomaterials by combining noble metals with other metals or oxides and using them in process analysis and quantitative detection.

## 1. Introduction

The electron distribution of nanoparticles will be disturbed by the electromagnetic field and vary from the original equilibrium position as a result of light irradiation, leading to repeating oscillations (Figure 1). Because it is confined to a finite volume of nanoparticles and cannot spread outward, this oscillation is known as local surface plasmon resonance (LSPR). The unique physical optical property of nanoparticles, particularly noble metal nanomaterials, is LSPR [1,2]. Because its scattering intensity and formant position are greatly influenced by material type, shape, and surrounding dielectric environment, it is frequently employed in biological sensing detecting technology. Simultaneously, in order to gain more sensitive sensing information of higher quality, sensor technology based on the LSPR properties of nanomaterials has been subjected to ever-increasing demands. A wide range of materials with various morphologies and components has arisen, resulting in numerous significant achievements in chemical and life science research [2,3]. Dark-field microscopy (DFM) is a useful tool for determining the spectral characteristics of nanomaterials. A dark-field sensing system is generated when nanomaterials mix with certain ligands to cause observable spectrum shifts, which can be used for process analysis and quantitative detection [4].

Plasmon research has made significant progress over the last two decades, with the development of advanced nanomaterials preparation methods, characterization techniques, and computing power and speed, and is now widely used in data storage, biosensing, optical imaging, photoelectric integration, and other fields [4,5,6,7]. Simultaneously, the advancement of spectrograph and detector (camera) technology has brought in a new era of dark-field microscopy and plasmonic nanomaterials spectroscopy [8,9]. The LSPR properties of noble metal nanoparticles are heavily dependent on their composition, shape, and surface dielectric constant, providing a useful detection platform for biochemical sensing technologies, particularly in the field of biochemical analysis. Furthermore, the LSPR capabilities of nanoparticles are commonly exploited in several enhanced surface optics processes, such as plasmon resonance energy transfer, fluorescence amplification, and surface-enhanced Raman scattering. The LSPR scattering spectra of ionic nanomaterials were employed to achieve molecule-level detection, while plasmonic nanomaterials were used to produce nanoscales for dynamic analysis of molecular structure changes [8]. In terms of material synthesis, notable advances in the technique of synthesizing nanoparticles with adjustable form and size have been made in recent years [3]. Many research groups, both at home and abroad, have achieved groundbreaking discoveries, including chiral nanoparticles, bimetallic co-composite materials, and non-noble metal plasmonic nanomaterials [2,4,10,11]. The photoelectric characteristics of nanoparticles are intimately related to the materials’ selection and structure. Even minimal structural variations have a substantial impact on the macroscopic physical properties, which, in turn, affect the device’s final performance. It is critical to investigate the material synthesis technique, nanostructure, and photoelectric properties of devices in order to acquire high-quality sensor parts and increase device performance and yield. At the moment, sensitive materials for sensor parts include single-component materials dominated by noble metal nanoparticles, as well as composites combining noble metal nanomaterials and other materials [3]. There is no doubt that the advancement of materials science opens up opportunities for the development of new sensor components.

This review will provide a thorough introduction to plasmonic nanomaterials based on dark-field sensing systems, including single-component materials and composite materials, as well as the impact of material composition, morphology, and structure on their plasmonic properties and applications. The future development of this hot field is forecasted towards the end of this study. It is worth noting that many excellent review articles summarizing plasmonic sensing research work have been published in recent years, including, but not limited to, dark-field sensing technology [4], plasmonic sensing and control of single-nanoparticle electrochemistry [12], and colorimetric detection and application based on LSPR properties of gold nanoparticles [13]. This paper primarily introduces the recent research and application advancements of plasmonic nanomaterials. We apologize for not being able to include all of the best work in this field due to space constraints.

## 2. LSPR and DFM

### 2.1. The Theory of LSPR

Conduction band electrons in metal nanostructures paired with electromagnetic fields ignite LSPRS. The aforementioned modes are directly excited by light under specific phase-matching conditions, and they typically exist towards the middle wavelength of the oscillating electromagnetic field. Local surface plasmons can also be excited in other nanostructures besides solid metal nanoparticles, such as nanocore shells with dielectric structures. Due to the resonance-enhanced absorption and scattering, which occurs in the visible range for gold and silver nanoparticles, the particles exhibit vivid colors in transmission and reflection.

When the radius a of a nanoparticle is much smaller than the wavelength λ of light in the surrounding medium, the interaction between the electromagnetic field and the nanoparticle can be analyzed using a simplified quasi-static approximation method. Under such conditions, the phase of the simply resonant electromagnetic field can be considered to be constant for the nanoparticles, so the problem can be transformed in the case of metal nanoparticles in a uniform electrostatic field, and the spatial magnetic field distribution can be calculated. First, the simplest geometric analytic method can be used (Figure 2), such that an isotropic uniform sphere of radius a is located in the standard electrostatic field $\vec{E}=E_{0}\hat{z}$, where $\varepsilon_{m}$ is the permittivity of the dielectric environment in which the metal nanoparticles are located, and where the electric field lines are parallel to the z direction away from the sphere. The dielectric response of a sphere is determined by the dielectric function and is generally reduced to the complex constant ε.

By solving the Laplace equation and based on appropriate boundary conditions, the polarizability of metal nanospheres can be obtained as shown in Equation (1):(1)$$\alpha =4\pi a^{3}\frac{\varepsilon -\varepsilon_{m}}{\varepsilon +2\varepsilon_{m}},$$

Equation (1) is the core result of the excitation of isoplasmons on a local surface. It is possible to calculate the polarization intensity of a spheroid with an electrostatic approximation of a mid-wavelength diameter. It is noteworthy that this has the same form as the Clausius–Clapeyron function. For further understanding, please refer to Jackson’s classic monograph on electrodynamics [14].

For a small sphere with a radius much smaller than the wavelength of light, it can be equivalent to an ideal dipole under the quasi-static approximation, which is suitable for a time-varying field and ignores the spatial delay effect caused by particle size. Under the irradiation of a plane wave, the oscillating dipole moment induced by an electric field is related to the polarizability, and the dipole radiation leads to the scattering of a spherical wave, which can be regarded as the point dipole radiation. For a detailed description of dipole radiation properties, refer to the textbook on electromagnetic field theory [14]. From an optical point of view, an interesting feature of resonance enhanced polarization is the enhanced light scattering and absorption of metal nanoparticles. The corresponding absorption cross sections $C_{abs}$, scattering cross sections $C_{sca}$, and extinction cross sections $C_{ext}$ can be derived from the electromagnetic field theory related to oscillating electric dipoles and the Poynting Vector. The expressions are as follows:(2)Cabs=kIm{α}=4kπa3Im{ε−εmε+2εm}
(3)$$C_{sca}=\frac{k^{4}}{6\pi }{\left|\alpha \right|}^{2}=\frac{8}{3}k^{4}\pi a^{6}{\left|\frac{\varepsilon -\varepsilon_{m}}{\varepsilon +2\varepsilon_{m}}\right|}^{2}$$
(4)$$C_{ext}=C_{abs}+C_{sca},$$
where k is the wave number of the incident light. For small particles with a radius much smaller than the wavelength of light, the absorption rate greatly exceeds the scattering rate. The above formula is not only applicable to metal nanomaterials but also to other scattering media.

In addition to spherical nanoparticle shapes, the basic physical description of local surface plasmon resonance of subwavelength metal nanostructures is also applicable to other situations. For example, under the condition of quasi-static approximation, for the ellipsoid with semi-axis $a_{1}\leq a_{2}\leq a_{3}$, as shown in Figure 3a, $\frac{x^{2}}{a_{1}^{2}}+\frac{y^{2}}{a_{2}^{2}}+\frac{z^{2}}{a_{3}^{2}}=1$ is satisfied, and the expression of polarization $\alpha_{i}\;(i=1,\;2,\;3)$ with spindle is shown in Equation (5):(5)$$\alpha_{i}=4\pi a_{1}a_{2}a_{3}\frac{\varepsilon (\omega )-\varepsilon_{m}}{3\varepsilon_{m}+3L_{i}(\varepsilon (\omega )-\varepsilon_{m})},$$

$L_{i}$ is the geometric factor, expressed as (6):(6)$$L_{i}=\frac{a_{1}a_{2}a_{3}}{2}\int_{0}^{\infty }\frac{dq}{\left(a_{i}^{2}+q)f(q\right)},$$
where $f(q)=\sqrt{\left(q+a_{1}^{2})(q+a_{2}^{2})(q+a_{3}^{2}\right)}$, the geometric factor is satisfied $\sum_{}^{}L_{i}=1$, and the sphere is satisfied $L_{1}=L_{2}=L_{3}=\frac{1}{3}$. As an alternative approach, the polarizability of the ellipsoid is usually expressed by the depolarizing factor $\tilde{L_{i}}$, defined by $E_{1i}=E_{01}-\tilde{L_{i}}P_{1i}$, where $E_{1i}$ and $P_{1i}$ are the intensity of the electric field $E_{01}$ inside the particle along the principal axis i and the induced polarization electric field, respectively. The conversion relationship between $\tilde{L_{i}}$ and L is shown in Equation (7):(7)$$\tilde{L_{i}}=\frac{\varepsilon -\varepsilon_{m}}{\varepsilon -1}\frac{L_{i}}{\varepsilon_{0}\varepsilon_{m}}.$$

For a long ellipsoid, the two short axes are equal $a_{2}=a_{3}$, while for a flat ellipsoid, the two long axes are equal $a_{1}=a_{2}$. Equation (5) shows that the ellipsoidal metal nanoparticles have two spectral separation plasmonic resonances, and the conduction electrons oscillate along the long and short axes, respectively. In the case of oscillating along the long axis, the spectrum is significantly red-shifted compared to the plasmon resonance on the sphere of the same volume. Based on this, metal nanoparticles with a large length–diameter ratio can be used to achieve plasmon resonance in the near infrared region of lower frequency in the experiment. It is worth noting that the long axis must also be significantly smaller than the wavelength of the excitation light to be effective.

Figure 3b shows the comparison between the resonant wavelength predicted by Equation (5) and that calculated by quasi-static approximate simulation. As can be seen in the figure, the two curves match very well.

Similar analytical methods are also applicable to spherical or ellipsoidal microstructures coated with different materials. For a core–shell structure whose inner diameter is $a_{1}$, dielectric function is $\varepsilon_{1}(\omega )$, outer diameter is $a_{2}$, and dielectric function is $\varepsilon_{2}(\omega )$, its polarizability can be expressed as in Equation (8):(8)$$\alpha =4\pi a_{2}^{3}\frac{\left(\varepsilon_{2}-\varepsilon_{m})(\varepsilon_{1}+2\varepsilon_{2})+f(\varepsilon_{1}-\varepsilon_{m})(\varepsilon_{2}+2\varepsilon_{2}\right)}{\left(\varepsilon_{2}+2\varepsilon_{m})(\varepsilon_{1}+2\varepsilon_{m})+f(2\varepsilon_{2}-2\varepsilon_{m})(\varepsilon_{1}-\varepsilon_{2}\right)},$$
where $f=a_{1}^{3}/a_{2}^{3}$ is the volume fraction of the total particle volume in the interior of the sphere [15].

In addition to the above standard model of subwavelength metal nanoparticles, the classical theory is also the Mie theory proposed by German physicist Gustav Mie, which is applicable to the absorption and scattering of electromagnetic radiation based on spheres of arbitrary size in uniform media. In addition, the Gan model expanded from Mie theory is applicable to the model of a columnar structure, and the relationship between the length–diameter ratio and resonance wavelength of columnar nanoparticles is obtained [15].

### 2.2. DFM

Plasmonic nanosensors generally consist of three components, which are the molecular recognition part (specific ligand), the signal converter (plasmonic nanoprobe), and the detection instrument. The experimental principle is as follows: molecular recognition leads to a change in the surface refractive index of the plasmonic nanoprobe or produces electromagnetic coupling between particles, causing a shift in the LSPR spectrum and even visible color changes. In fact, utilizing standard spectrometers to obtain the spectrum of nanoparticles is relatively convenient. The advantage of utilizing DFM is that it allows us to see how the spectrum of a single nanoparticle changes. Clearly, the application of DFM can improve sensitivity and LOD, as well as analyze the chemical binding process by changing the spectrum of single-particle nanoparticles.

Plasmonic nanoparticles have been widely used in surface-enhanced Raman scattering (SERS) imaging and dark-field localized plasmon resonance (LSPR) imaging due to their special nano optical properties. SERS has ultra-high sensitivity and can detect liquid samples with ultra-low concentrations. However, due to its lack of quantitative detection capability and the fact that only a few basic metal particles with roughened surfaces (such as gold, silver, copper, lithium, sodium, and so on) can create SERS, the application range of SERS imaging is limited. Although various quantitative SERS analysis methods have been developed in the laboratory, such as adding internal standards of known concentrations to the determinand, these methods have achieved excellent linear curves in small ranges but have also made detection more difficult [16,17]. In addition to quantitative detection, DFM is better than SERS in exploring mechanisms during the reaction process [4].

In this section, we introduce the dark-field microscopic imaging system, a powerful tool for collecting LSPR scattering spectra of nanoparticles.

#### 2.2.1. Mechanical Scanning Dark-Field Microscopic Imaging System

The main structure of a mechanical scanning dark-field microscopic imaging system is a transmitted dark-field microscope. The incident light cannot be directly irradiated on the sample due to the effect of a dark-field condenser, and only the scattered light of nanoparticles enters the objective lens. The scattered light passes through a slit and through a grating and is collected by the camera. It has the following three characteristics: First, the incident light must be white. Second, to collect the scattering spectrum of the target nanoparticle, the slit must first be moved directly above the target nanoparticle. Third, the spectrum of the particles is calculated by the gray value projected on different pixels after splitting by the grating. As shown in Figure 4, colorful scattered light spots in the dark field can be observed through the eyepiece. When the scattering spectrum of a particle needs to be located, the slit is moved above the nanoparticle. Figure 4 shows an example of the application of the mechanically scanned dark-field microscopic imaging system [18]. The researcher used the slit to precisely locate the scattering spectrum of a single nanoparticle and realized the specific detection of mercury ions by using the spectral changes after the adsorption of mercury ions by the ligand probe on the nanoparticle surface.

Although the mechanical scanning dark-field microscopic imaging system was the earliest such system, it still has an irreplaceable position today. First of all, it has high detection accuracy. To reconstruct the scattering spectrum of nanoparticles, each data point of the corresponding wavelength needs to be obtained, and the final spectrum can be synthesized by connecting the points into lines. Secondly, the system can effectively avoid the influence of stray light. Especially in some special application scenarios, such as the measurement of semiconductor nanoparticles, some special particles themselves will produce strong fluorescence due to high-energy photon excitation, while the mechanical scanning dark-field system will not be affected by stray light due to the existence of slits. Its disadvantages are as follows: First, the system requires a high precision nano translation platform, a high performance CCD, the use of dual cameras, as well as an autofocus objective lens and other sophisticated components. The system is difficult to develop and the cost is relatively high, making it unsuitable for popularization and application. Secondly, the detection needs positioning, translation, exposure, and other operations; these steps are more complicated, resulting in the slow detection speed of the system. In addition, it is difficult to collect scattering spectra of multiple particles at the same time because of the existence of the slit.

#### 2.2.2. Dark-Field Microscopic Imaging System with Incident Wavelength Modulation

The dark-field microscopic imaging system with incident wavelength modulation is a new type of system that has emerged in the last decade, as shown in Figure 5 [19]. One feature is that the incident light is treated by spectrophotometry before entering the dark-field condenser. At a certain time, only light of a specific wavelength (such as 400 nm) enters the dark-field condenser and irradiates the sample. The scattered light from the nanoparticles is collected by the camera, resulting in a dark-field image at an incident wavelength of 400 nm. An image set can be obtained by continuously scanning the wavelength of the incident light; for example, from blue light to red light, as shown in Figure 5b. The scattered spectrum of each particle can be obtained from this image set. The dark-field microscopic imaging system with incident wavelength modulation has the following characteristics: First, incident light is monochromatic light. Second, the beam splitting and scanning must be precisely synchronized. Third, compared with a single image produced by a mechanical scanning dark-field microscopic imaging system, the dark-field image data obtained by the system comprise an image data set. The scattered spectrum of a single nanoparticle needs to be extracted centrally from the entire image, as shown in Figure 5c. Each data point on the scattering spectrum is derived from the processing result of one of the dark-field images, and the wavelength resolution is related to the number of images.

It can be seen that this system has obvious advantages: First, because the system can obtain the scattering spectra of a large number of nanoparticles (more than thousands of nanoparticles) at the same time, it can be used for high-throughput detection and improve the reliability of the experiment. Second, the detection speed is fast. Depending on the set scanning speed and camera frame rate, the system can quickly obtain the scattered spectrum of thousands of nanoparticles, usually in tens of seconds. The characteristics of fast detection can meet the needs of many detection applications. Of course, the system also has some limitations. The first is that it requires more sophisticated data processing software. In addition, due to the need for wavelength scanning to obtain the scattered spectrum, it is not suitable for studying the time-varying spectrum, and it is difficult for the system to detect spectral changes within a few seconds or even tens of seconds.

#### 2.2.3. Summary of DFM

It is difficult to say which of the above two dark-field microscopic imaging systems is better. Researchers can build a suitable system according to the research object and application field. For example, the mechanical scanning dark-field microscopic imaging system is needed to study the transfer of charge process on the surface of nanoparticles, perform catalytic analysis, and enable nanoscale construction to dynamically study the change of molecular structure. For biochemical detection and quantitative analysis, it is necessary to study the spectral changes of a large number of nanoparticles in order to obtain reliable detection data. At this time, the incident wavelength modulated dark-field microscopic imaging system has obvious advantages.

## 3. Materials and Applications

In the past two decades, a vast number of plasmonic nanomaterials with diverse shapes and compositions have been synthetized and thoroughly investigated. Notably, gold and silver nanoparticles have received significant attention due to their remarkable LSPR properties in the visible region. With respect to biochemical detection, gold nanoparticles outperform other noble metal nanoparticles by benefiting from their non-toxicity and resistance to oxidation as inert metal materials. Additionally, they are easily synthesizable and modifiable, and exhibit high biocompatibility. Similarly, silver nanoparticles have high reactivity and scattering strength, which make them ideal for electrochemical oxidation-reduction imaging and process tracking. In recent years, the study of single nanoparticles has rapidly grown across different scientific fields. Here, we will specifically focus on the applications of the LSPR properties of nanoparticles in the realm of biochemical sensing.

For both gold and silver nanoprobes, resonance is concentrated in the visible region, resulting in bright colors owing to resonance-enhanced absorption and scattering. Scattered light exhibits high intensity. Single-particle optical observation under DFM can detect silver nanoparticles as small as 20 nm and gold nanoparticles as small as 30 nm in diameter. Sensing strategies can be mainly categorized into the following three approaches. The first sensing approach is based on changes in the dielectric environment around nanoparticles, such as direct adsorption of test objects on single-particle nanospheres and nanorods, leading to alterations in the plasmonic dielectric environment and scattering spectrum. The advantage of this sensing technique is that the plasmonic system is simple to prepare and exhibits high experimental repeatability. However, it is easily interfered with by other adsorbable substances, and wavelength offsets are usually less than twenty nanometers, requiring high-precision DFM equipment. The second approach relies on the electromagnetic coupling caused by the aggregation or depolymerization of nanoparticles, which leads to the wavelength offset; for instance, the formation of dimer or core–satellite structures under the influence of the test object, or the destruction of dimer or core–satellite structures under the influence of the test object. The advantage of this sensing technique is its high sensitivity, and the spectral offset is typically more than 30 nanometers. However, the preparation of the plasmonic probe modification process required is usually complex. The third sensing approach is based on the transfer of charge on the surface of nanoparticles, which is frequently applied to monitor the catalytic process, leading to changes in spectral and scattering intensity. This sensing technique has the advantage of high sensitivity but exhibits the disadvantage of poor reproducibility.

In addition, combining single-component noble metal nanoparticles, such as gold and silver, with other nanomaterials to construct composite nanoparticles with special structures can impart unique optical properties to the nanomaterial. This method is useful for enhancing the LSPR performance of plasmonic nanoparticles. This article mainly explores different shapes and materials of plasmonic nanoprobes in dark-field sensing systems, including single-component and multi-component plasmonic probes. This review will classify related research according to the material of plasmonic probes. In this review, single-component refers to a plasmonic nanoprobe composed of a single material, and multi-component refers to a plasmonic nanoprobe composed of two or more materials.

### 3.1. Single-Component Plasmonic Nanomaterial

After modification with specific ligands, plasmonic probes can specifically bind to detection targets and use changes in the surrounding dielectric environment or the electromagnetic coupling between nanoparticles for sensing. In addition, the optical properties of plasmonic probes can be adjusted by changing their material, size, and nanoparticle spacing. Gold, silver, and copper nanoparticles are the most commonly used materials for plasmonic probes. Sekhon et al. determined the refractive index sensitivity and quality factor of gold, silver, and copper nanoparticles by changing the nanoparticle size, shape, and surrounding refractive index [20]. Figure 6a,b show the plasmonic resonance wavelength of spherical and rod-shaped Au, Ag, and Cu nanoparticles as a function of diameter and aspect ratio. As the diameter and aspect ratio of spherical and rod-shaped Au, Ag, and Cu nanoparticles increase, their resonance wavelength shifts to the red light. Figure 6c,d show their specific LSPR performance. As the diameter of Au, Ag, and Cu spherical nanoparticles increases, their refractive index sensitivity increases, and their full width at half maximum (FWHM) also increases. For rod-shaped nanoparticles, as shown in Figure 6e,f, the linear degree of Ag nanoparticle refractive index sensitivity increases with the increase in aspect ratio, and the FWHM is relatively lower. Therefore, compared to Au and Cu nanoparticles, Ag nanoparticles exhibit better LSPR sensing performance. However, the oxidation problem of Ag in solvents must be considered, while Au nanoparticles are relatively stable, so choosing the appropriate material as the plasmonic probe based on the application is the optimal solution.

#### 3.1.1. Nanospheres

Nanospheres (NPs) have isotropic characteristics. Under the DFM, the LSPR spectrum of a single nanosphere (NP) is independent of its orientation. Due to their simple preparation method and high yield, NPs have become one of the most commonly used nanomaterials for researchers. Liu et al. designed a functionalized plasmonic for Cu^2+^ detection by immobilizing a Cu^2+^-specific DNA enzyme probe on the surface of 50 nm AuNPs, using an aggregation strategy [21]. As shown in Figure 7, the plasmonic forms dimers when AuNPs are added in the absence of Cu^2+^, but no aggregation occurs when Cu^2+^ is present. The limit of detection(LOD) for Cu^2+^ reaches 0.082 nM, and the dynamic range is 0.1 nM–5 µM. In its linear range, the LSPR scattering spectra show a shift of several nanometers to 20 nm.

Gai et al. utilized 70 nm AuNPs as plasmonic sensing elements to detect Hg^2+^ in a solution [22]. Upon the presence of Hg^2+^ in the sample, the oligonucleotides immobilized on the surface of AuNPs undergo hybridization, leading to the formation of dimers and a corresponding color change from green to yellow in the scattering spectra under the DFM. Quantitative analysis of Hg^2+^ was achieved by analyzing the color change in dark-field imaging. The limit of detection (LOD) and dynamic range of this method were 1.4 pM and 0.005–25.0 nM, respectively.

Guo et al. designed three strategies using an aggregation strategy on 50 nm AuNPs, as shown in Figure 8a: (1) Two functionalized AuNPs form a dimer under the detection of Cu^2+^, resulting in a red shift of 60 nm (550 nm to 610 nm) in the scattering spectra under DFM; (2) two double-modified AuNPs are synthesized, and a dimer is formed under the action of azide/thiol-modified DNA and alkyne/anti-HER2 in the presence of Cu^2+^, as shown in Figure 8b. This method was analyzed by the ratio of red channel intensity to green channel intensity (R/G) of the imaging picture, showing a linear range of 10^−13^–10^−8^ M; (3) the second strategy is coupled with rolling circle amplification (RCA) to detect HER2 or HER2 overexpressing cancer cells, as shown in Figure 8c. The maximum spectral red shift under DFM was up to 680 nm, which was 70 nm higher than the first strategy [23].

Xiao et al. utilized Au-S bonds to immobilize mercaptophenylboronic acid (MPBA) on the surface of AuNPs [24]. MPBA can form a stable boronate ester bond with the molecule of D-galactose. In the presence of D-galactose, the functionalized AuNPs undergo aggregation and dimerization, resulting in a red shift of the LSPR scattering spectra from green to yellow under DFM. Moreover, the research group prepared AuNPs of different sizes (25 nm, 35 nm, 45 nm, and 50 nm) using the seed growth method and simulated the LSPR spectra and detection of single nanoparticles and dimerized nanoparticles under DFM, as shown in Figure 9. After a comparative analysis, they found that 45 nm AuNPs were the most suitable for camera detection due to their peak wavelength and intensity. The LOD for the boronic-acid-functionalized AuNPs towards D-galactose was 0.83 nM, with an LSPR red shift of approximately 55 nm.

In addition to biological and chemical sensing, the emergence of single-nanoparticle analysis and DFM imaging technology also aids in the visual monitoring of the dynamic process of practical photocatalytic reactions (PPR). This can be demonstrated by real-time monitoring of the optical behavior of individual plasmonic nanoparticles during the reaction process. Mulvaney’s group first discovered that the change in electron density in metal nanoparticles caused by injecting or extracting electrons from them may lead to a blue or red shift of the LSPR spectral band, indicating that LSPR spectra can be used to measure the concentration of transferred electrons in real-time and quantitatively evaluate the metal catalytic activity of oxidation reduction [25,26]. Mulvaney’s findings laid a solid foundation for tracing the electron transfer pathway and real-time measuring of the electron gain and loss rate of individual particles.

Yuan et al. studied the proton-coupled electron transfer (PCET) process of 4-nitrothiophenol (4-NTP) conversion to 4-dimethylaminoazobenzene (DMAB) catalyzed by 40 nm AgNPs driven by LSPR in plasmon-driven photocatalytic reactions (PPR) [27]. They measured the real-time electron density and electron gain and loss rate, and visually monitored the electron transfer dynamics on a single AgNP. The results showed that PPR could be dynamically visualized, and the changes in scattering intensity and wavelength shift in the LSPR scattering spectra during this period indicated that PPR was essentially due to PCET.

For plasmonic nanospheres, single-particle optical observation under DFM can detect silver nanoparticles as small as 20 nm and gold nanoparticles as small as 30 nm in diameter. According to the simulation and experimental verification, larger particle sizes typically exhibit higher peak shifts and peak intensities. However, increasing the particle size may cause the line broadening of the plasmon resonance peak, leading to the limitation of spectral resolution for wavelength changes. In other words, the increase in spectral full width at half maximum (FWHM) has a negative impact on sensor sensitivity. Therefore, choosing an appropriate nanoparticle size can effectively improve the detection sensitivity under DFM. Current research shows that AuNPs or AgNPs with a diameter of 40–80 nm are the optimal choice for spherical metal plasmonic nanoparticles for biochemical sensors.

#### 3.1.2. Nanorods

Theoretical studies on the surface plasmon resonance conditions of rod-shaped plasmonic nanoparticles have shown that, under the condition of fixed materials, the spectral sensitivity—which refers to the relative shift of the resonance wavelength with respect to the change in the refractive index of the surrounding material—is mainly related to the size and aspect ratio of the nanorod [28]. On the other hand, the spectral sensitivity of spherical plasmonic nanoparticles is only related to their size. Therefore, in some applications, rod-shaped plasmonic nanoparticles have more advantages than spherical plasmonic nanoparticles. Due to their shape tunability, rod-shaped plasmonic nanoparticles can control their absorption and scattering characteristics by adjusting their aspect ratio. Additionally, there has been much research on the assembly and topological structures of rod-shaped surface plasmon resonances, such as the assembly of individual nanorods and the construction of nanowire networks. Thus, rod-shaped surface plasmon resonances have great potential in areas such as biosensing, molecular recognition, catalysis, and optoelectronics compared to their spherical counterparts.

Sim et al. designed plasmonic Au nanorods (AuNRs) with a direct adsorption strategy to detect p53 protein in the growth arrest and DNA damage-inducible 45 (GADD45) promoter and evaluate its association with breast cancer [29]. The AuNRs in this plasmonic sensing platform were synthesized using seed-mediated growth, with an aspect ratio of 3.6 (length: 107.22 ± 3.64 nm; width: 29.48 ± 1.06 nm). The specific probe was immobilized onto the surface of the gold nanorods, and detection of the p53 protein was carried out using dark-field microscopy. The scattering spectrum exhibited up to a 20.9 nm red shift, with a linear range of 10–106 fM and an LOD of 11.47 fM.

Brimmo et al. used a direct adsorption strategy and a AuNR plasmonic sensing platform to perform in situ and high-throughput detection of cytokines related to adipose tissue inflammation via DFM [30]. This plasmonic sensing platform was prepared using cetyltrimethylammonium bromide (CTAB) to coat AuNPs with an aspect ratio of 80 ± 5 nm in length and 40 ± 3 nm in width. Multiple cytokines, such as tumor necrosis factor-α (TNF-α), interleukin-6 (IL-6), interleukin-10 (IL-10), and interleukin-4 (IL-4), were detected, and the LSPR scattering spectra of the AuNRs shifted from 662 nm to 681 nm under DFM. The sensitivity of this detection method reached 20 pg/mL^−1^.

Huang et al. used AuNRs with a size of around 79 × 30 nm as a plasmonic sensing platform for detecting (microRNA) miRNA-21 [31]. As shown in Figure 10, the plasmonic sensing platform was based on the displacement reaction between the target miRNA-21 and the probe, which produced H_2_O_2_. I^−^ was used to etch the AuNRs plasmonic sensing platform with the catalytic effect of H_2_O_2_. The scattering intensity of the AuNRs was significantly reduced under DFM, and the change in scattering intensity was correlated with the concentration of miRNA-21, allowing for quantitative detection. The LOD reached 71.22 fM.

Ahijado-Guzmán et al. extended the application of AuNRs by observing the LSPR scattering curves of AuNRs of different sizes to determine the optimal induction distance of AuNRs, providing information on the distance between plasmonic sensing platform and protein interactions. Based on this observation, they found that cardiolipin (CL) and local membrane curvature strongly influenced the minimal protein oscillation mechanism [32].

Luo et al. designed a plasmonic sensing platform based on AuNRs, and by using an aggregating strategy, they detected miRNA-21 and miRNA Let-7a using DFM [33]. The plasmonic sensing platform was prepared by functionalizing the surface of the gold nanorods with hairpin DNA with an aspect ratio of approximately 2:1 (100 nm in length and 50 nm in width). AuNRs were assembled into dimers in two ways, end to end and parallel, in the presence of miRNA-21 and miRNA Let-7a. Under DFM, the LSPR spectra of single AuNRs and both types of dimers were characterized at different angles, as shown in Figure 11. End-to-end dimers showed twice the aspect ratio of a single AuNR, while parallel dimers showed half the aspect ratio of a single AuNR. As a result, when the polarization direction of the incident light was different from the direction of the long and short axes of the plasmonic sensing platform, different spectral characteristics were observed. By imaging and analyzing end-to-end dimers and parallel dimers under DFM, Luo et al. successfully achieved the quantitative detection of miRNA-21 and miRNA Let-7a in serum, with LODs of 1.72 fM and 0.53 fM, respectively.

Tan et al. also used the same aggregating strategy to detect target DNA. The AuNRs they used had the same aspect ratio, approximately 2:1, as Luo et al. (51.7 nm in length and 24.3 nm in width) [34]. While the characteristics of the probes for AuNR surface modification were different for different target molecules, Tan et al. also characterized the dimers formed by both assembly methods. As shown in Figure 12, the effect of different inter-AuNR distances on the LSPR scattering efficiency was characterized at different incident polarization directions for single AuNRs and dimers assembled in end-to-end and parallel ways. The LSPR scattering efficiency of both assembly modes increased when the polarization direction was perpendicular to the long axis, with no significant spectral shift. However, when the polarization direction was perpendicular to the short axis, the spectral shift of the scattered light of the dimers increased with a decrease in the inter-AuNR distance, providing a good theoretical basis for the study of plasmonic sensing platforms based on AuNR aggregation strategies.

Ha et al. analyzed the curvature of the LSPR spectra of AuNRs with an aspect ratio of 2.92 (25 nm × 73 nm), both in a bare state and in the presence of pyridine, as shown in Figure 13a–c, and found that the inflection point of the spectra corresponded to the maximum sensitivity of the LSPR effect [35]. Interestingly, the inflection point on the low−energy side, or the inflection point on the longer−wavelength side, exhibited higher sensitivity and greater wavelength shifts in practical measurements, as shown in Figure 13d, which may be related to chemical interface damping (CID). Another interesting technique of this group is to etch AuNR (25 nm × 73 nm) and analyze the variation of LSPR spectral linewidth after different oxygen plasmonic etching [36]. As shown in Figure 14, the linewidth of the AuNR changed little in the first 180 s because the CTAB on the surface was removed, and the AuNR shape did not change significantly. However, the size and shape of the AuNR changed significantly after 180 s, with the length decreasing significantly from 64.3 nm (120 s) to 61.6 nm (180 s), while the transverse length increased from 22.5 nm (120 s) to 25.3 nm (180 s). The linewidth of AuNR increases with the increase in etching time, which can be attributed to the decrease in the length−to−diameter ratio and the increase in LSPR damping.

#### 3.1.3. Nanoplate

Due to the sharp edges of the polygonal nanosheets and their enhanced electric fields, polygonal nanosheets exhibit strong LSPR scattering and show high−intensity, narrow−distribution characteristics under DFM. Therefore, the use of their unique edge effects to achieve material detection is also a hot research topic.

Gao et al. designed a plasmonic system of gold triangular nanoplates (AuNPLs) and detected the inhibitory effect of pyrophosphate (PPi) on the etching of Cu^2+^ and I^−^ in AuNPLs [37]. As shown in Figure 15, Cu^2+^ and I^−^ undergo an oxidation reaction to produce I2, which etches the AuNPLs with a side length of about 65 nm to produce Au^+^. The plasmonic system changes from a triangle to a circle, and its LSPR scattering light undergoes a blue shift of 50 nm (from 625 nm to 575 nm) under DFM. When PPi is present, it reacts preferentially with Cu^2+^ instead of I^−^, inhibiting the production of I2. As a result, the plasmonic LSPR scattering spectra do not change because etching cannot occur. The LOD for PPi reached 1.09 nM, and the linear range was 7−100 nM using this method.

Huang et al. designed a plasmonic system of gold hexagonal nanoplates (AuHNPs) based on the edge effect of the nanoplates and successfully monitored the real−time formation process of Au@Hg nanocomposites [38]. They found that when the distance between the opposite sides of AuHNPs was greater than 500 nm, their scattering images showed a donut−shaped far−field scattering pattern (DNSP) due to the edge effect, as shown in Figure 16a. Meanwhile, Huang et al. characterized the LSPR spectra of AuHNPs within a range of 500−1000 nm for opposite−side distances: the scattering wavelength was always located at 640 nm and independent of the size; the scattering intensity increased with increasing size; and the degree of peak separation was linearly related to the size, as shown in Figure 16b. Therefore, the etching process of AuHNPs can be monitored in real time by the distance between the dual peaks in their LSPR imaging, which reflects their size.

In theory, LSPR can be excited in any semiconductor, metal, or alloy material that has a small imaginary dielectric constant and a large negative real dielectric constant [39]. In addition to gold and silver nanoparticles, LSPR phenomena can be excited in other metal and/or metal oxide nanocomposites. For example, George C. Schatz synthesized triangular aluminum nanoparticles using nanosphere lithography (NSL) and studied their LSPR properties. The results showed that the presence of an Al_2_O_3_ oxide layer, especially at the tip of the triangle, caused a significant red shift in the LSPR λmax. Compared to similarly sized and geometrically shaped unoxidized triangular nanoparticles, the oxide layer significantly reduced their refractive sensitivity in solvents. Comparison of Al, Ag, Cu, and Au triangular nanoparticles of similar size and geometric shape revealed that the LSPR λmax had the following characteristics: Au > Cu > Ag > Al, while the linewidth satisfied Al > Au > Ag > Cu. This provides a good data basis for research on Al-related plasmonic materials.

Kristensen et al. designed nano-pores on the surface of an aluminum plate with different structural periods, and the combination of the nano-pore structure and LSPR properties of the disk achieved angle-insensitive bright colors [40]. The scattering light color of the aluminum nanoplates can be controlled by the structural period size of the nano-pores. The aluminum nanoplates were prepared by depositing them on a mother mold made in silicon by using electron beam lithography (EBL) and dry etching. The thickness of the plate was 80 nm, and the height of the nano-pores was 50 nm. The impact of the structural period on the resonance position was not significant because it only affected the mutual coupling between adjacent resonators. However, the nearest coupling did affect the extent to which changes in diameter affect the resonance position. For coating samples with a structural period of Λ = 160 nm, increasing the diameter of nano-pores by 1 nm led to a red shift of about 7 nm, while for Λ = 240 nm, it only resulted in a red shift of about 4 nm. This study has a large-scale potential for designing structural colors in everyday plastic products.

The resonance wavelength of the long axis of nanorods is connected to the nanorods’ length-to-diameter ratio, and the higher the ratio of length to diameter, the larger the resonance wavelength. In nanorod applications, the length–diameter ratio and particle size of nanorods are often chosen based on the best performance region of DFM. In general, nanorods with a length–diameter ratio of 2.5–3 are more typically utilized, and high optical imaging effects and detection sensitivity may be attained at this time.

#### 3.1.4. Core–Satellite Structure

Close-range particle coupling can greatly enhance the local field and meet the high sensitivity requirements for single-particle-level analysis. Compared to a single-particle plasmonic, a multi-particle coupled plasmonic shows a significant increase in spectral shift when aggregation or disaggregation occurs [41]. Therefore, the core–satellite structure is an effective design for improving sensitivity and lowering detection limits. Sönnichsen et al. designed a plasmonic by assembling 20 nm AuNP satellites onto a 60 nm AuNP core using hydroxylamine-mediated assembly [42]. The plasmonic was encapsulated with citrate salts for satellites and functionalized with methoxy-polyethylene glycol for cores. Hydroxylamine was then added to neutralize the charge of the citrate-coated spheres, causing them to partially lose their colloidal stability and subsequently adsorb onto the methoxy–polyethylene glycol-functionalized core spheres. By measuring the sensitivity of the core–satellite structure plasmonic and same-sized single-particle plasmonic, the superiority of the core–satellite structure in sensing performance over the single-particle plasmonic of the same material and volume was verified.

Gooding et al. designed a plasmonic using 67 nm and 10 nm AuNPs, which aggregated into a core–satellite structure under the action of the target analyte Interleukin 6 (IL-6) and achieved quantitative detection of IL-6, as shown in Figure 17 [43]. The monoclonal antibody and polyclonal antibody of IL-6 were respectively modified on the surfaces of 67 nm and 10 nm AuNPs, and they successfully aggregated into a core–satellite structure under the action of IL-6. Under DFM with mediocre performance, AuNPs smaller than 20 nm cannot be effectively detected. Therefore, the change in the scattering light color of the plasmonic was only related to the number of satellites bound on their surfaces. Based on this, the scheme successfully achieved the quantitative detection of IL-6 with a limit of detection (LOD) of 0.01 ng/mL.

Wei et al. designed a plasmonic using 50 nm and 8 nm AuNPs for detecting poly (ADP-ribose) polymerase-1 (PARP1) [44]. They modified specific active double-stranded DNA (dsDNA) on the surface of 50 nm AuNPs. When both nicotinamide adenine dinucleotide (NAD) and PARP1 were present, dsDNA cleaved NAD. The 8 nm AuNPs were attached to the 50 nm AuNPs through the dsDNA@PAR structure, forming a core–satellite structure. Under DFM, the LSPR scattering spectrum showed a maximum red shift of 83 nm, and the linear detection range was 0.2 to 10 mU.

Wang et al. prepared core–satellite nanostructures using 50 nm and 13 nm AuNPs as cores and satellites, respectively [41]. They modified the cores and satellites using short thiolated L-DNA and S-DNA, respectively. Through the gap-filling DNA extension and strand displacement hybridization triggered by telomerase, the dissociation of the core–satellite nanostructures took place, as shown in Figure 18. Under DFM, the LSPR scattering spectrum shifted from 625 nm to 560 nm, achieving the in situ quantitative detection of telomerase activity in cancer cell lines. The detection limit was as low as 1.3 × 10^−13^ IU.

Liu et al. designed a core–satellite-structured plasmonic combined with microfluidic technology to achieve in situ trace detection of Hg^2+^, as shown in Figure 19 [45]. The plasmonic consists of a AuNR with an aspect ratio of 2.5:1 (length: 37 nm; width: 10 nm) as a core, and a AuNP with a diameter of 10 nm as the satellite part. Under the action of the detection material Hg^2+^, the plasmonic is linked by oligonucleotides. Compared with a traditional spherical dimer plasmonic, this plasmonic has better sensitivity. It achieved a detection limit of 2.7 pM for Hg^2+^ in water. Table 1 shows the recent advances in the application of single-component plasmonic nanomaterials.

### 3.2. Plasmonic of Composite Materials

#### 3.2.1. Bimetallic Composite Plasmonic

Lin et al. prepared a Au@Ag core−shell−structured plasmonic by depositing Ag on the surface of 18 nm AuNPs [54]. The thickness of the Ag shell was 11 nm. As shown in Figure 20, Cd^2+^ induced the aggregation of the plasmonic, causing a new absorption peak to appear in the extinction spectrum of LSPR, and the difference between the original absorption peak and the new peak was 210 nm (407−617 nm). The increase in Cr^3+^ concentration resulted in an increase in the plasmonic’s absorption intensity. Through data analysis, quantitative detection of Cd^2+^ and Cr^3+^ in water was achieved under DFM, and their LODs were 11.5 and 26.8 nM, respectively. Xu et al. formed a core−shell−structured plasmonic composed of gold particles with Pt, Pd, Rh, PtPb, or PdPh and used it for monitoring the electrochemical deposition process under DFM [55].

In addition to the core−shell structure of ordinary NPs, other shapes of core−shell structures have also shown superior LSPR characteristics. Wang et al. designed a plasmonic probe of 50 nm gold nanoflowers (AuNFs), which exhibited a higher LSPR and stronger scattering wavelength shift than AuNPs [56]. As shown in Figure 21, alkaline phosphatase (ALP) catalyzes the hydrolysis of L−ascorbic acid 2−phosphate (AA2P) to generate L−ascorbic acid (AA). AA reduces Ag^+^ to deposit Ag on the surface of AuNFs, which changes the shape of AuNFs and causes a huge blue shift in the LSPR peak (90 nm, 655−565 nm). This achieves in situ detection without damage to ALP in cells; the LOD was 0.03 μU L.

Jiang et al. prepared a multifunctional therapeutic diagnostic plasmonic by co-loading AgNPs and hematoporphyrin monomethyl ether (HMME) into gold AuNFs [57]. When the plasmonic is engulfed by cancer cells, reactive oxygen species (ROS) in the cancer cells trigger the oxidation etching of AgNPs on the plasmonic, resulting in a spectral shift in the plasmonic’s scattering spectra. At the same time, ROS induce the activation of caspase-3 in cells, which cleaves the containing C-peptide caspase-3 specific recognition sequence (DEVD) and allows HMME to be released from the plasmonic, leading to significant fluorescence recovery associated with caspase-3 activity, as shown in Figure 22. This probe can be used for imaging of intracellular caspase-3 and ROS, distinguishing between cancer cells and normal cells. More importantly, the plasmonic showed excellent ability to kill cancer cells through the synergistic effects of chemotherapy and photodynamic therapy. The plasmonic effectively integrates the functions of promoting apoptosis and detection and can be used for cancer cell therapy and evaluation.

Huang et al. visualized the electroexchange (GE) reaction by monitoring the LSPR scattering spectra shift of Au, Ag, and an Au/Ag alloy under DFM [58]. As shown in Figure 23, when AgNPs undergo GE reaction with Au^3+^ in HAuCl_4_ solution, Ag is oxidized to Ag^+^ and dissolved in the solution, and Au^3+^ is reduced to Au and deposited on the surface of Ag, forming an Au/Ag alloy. During this process, the scattering spectra of the plasmonic shift significantly from blue to green, achieving the visualization of the GE reaction process.

Qu et al. designed a Au-covered Ag nanocage (NC)-structured plasmonic to achieve detection of miRNA-21 under DFM [59]. The plasmonic was spherical, with a diameter of 55 nm. miRNA-21 catalyzed the hairpin chains modified on the surface of the plasmonic to generate reactive oxygen species (ROS) and achieve the etching of the silver components of a Au/Ag NC. As the proportion of silver decreased, the scattering intensity of a single Au/Ag NC decreased rapidly, and the decrease rate was related to the content of miRNA-21. The linear range of the plasmonic probe for miRNA-21 detection was 0.1 fM to 10 fM.

#### 3.2.2. Other Multi-Component Nanomaterial Plasmonic Probes

Xu et al. designed a plasmonic probe with a Au@MnO_2_ core–shell structure with a size of 79 nm by depositing hydrothermally reduced MnO_2_ on the surface of 50 nm AuNPs [60]. They achieved quantitative detection of ALP in human serum. 2-phospho-l-ascorbic acid trisodium salt would decompose into l-ascorbic acid under the action of ALP. As a reducing agent, l-ascorbic acid triggered the decomposition of MnO_2_ on the surface of the plasmonic probe, and its LSPR scattering spectra under DFM could exhibit a maximum of 37 nm blue shift (from 582 nm to 545 nm). The dynamic range of the plasmonic probe for ALP detection was 0.06–2.48 mU/mL, and the LOD was 5.8 μU/mL. In addition, Xu et al. used the same plasmonic probe and sensing strategy to achieve glucose detection [61]. Based on the enzyme-catalyzed reaction of glucose oxidase (GOx) on glucose and the etching reaction of H_2_O_2_ on the surface of Au@MnO_2_ particles, the probe achieved quantitative measurement of glucose under DFM, with a linear range of 0.05–20 μM and an LOD of 12.9 nM.

Lin et al. designed Au@AgI core–shell nanorod plasmonic probes for H_2_S detection [62]. The probe had a AuNR core with a long axis of 35 nm and a short axis of 10 nm (aspect ratio of 3.5). The shell was made of Ag and reacted with I_2_ to form a AgI layer with a thickness of 2.1 nm. When exposed to H_2_S, the AgI shell rapidly transformed to Ag_2_S, causing a change in the LSPR scattering spectrum. H_2_S concentration was quantitatively analyzed using a colorimetry method, and the dynamic range of detection was 0.1–500 nM, with an LOD of 33 pM.

Zhao et al. designed an Au@FeOOH core–shell plasmonic probe for the detection of ALP under DFM, as shown in Figure 24 [63]. The probe was made of a gold core and a FeOOH shell. The reduction of ascorbic acid (AA) produced by the ALP-catalyzed hydrolysis of L-ascorbic acid 2-phosphate (AA2P) can etch the FeOOH shell on the surface of the probe, resulting in a blue shift of the scattering spectrum of the probe. The LOD for ALP was as low as 0.06 U/L.

Xu et al. designed core–shell plasmonic probes of Au@semiconductor(SC) using an electrochemical deposition method [64]. The CdS, CdSe, and ZnS were selectively deposited on the surface of AuNPs, forming a uniform shell with a clear metal/SC interface, which overcame a potential barrier caused by a large lattice mismatch between the two components, as shown in Figure 25. The formation process of the structure can be monitored in real time under DFM, and the extent of the LSPR scattering spectrum red shift characterizes the formation process of the plasmonic probe.

Huang et al. designed a shell-isolated plasmonic nanostructure (AgSHIN) with a core of AgNP and an ultra-thin shell of SiO_2_ [65]. This plasmonic probe initially had similar features to those of AgNP but exhibited higher chemical stability. By comparing with AgNP under DFM, it was determined that AgNP probes undergo weak oxidation in water, and the AgSHIN probes significantly improved the confidence level in imaging analysis compared with AgNP. Fritsch et al. used a polymer (poly-3,4-ethylenedioxythiophene) to modify the oxidation-reduction magnetic hydrodynamic (ORMHD) microfluidic system under DFM, enabling high-throughput single-nanoparticle (NP) in situ differentiation and manipulation in flowing mixtures through LSPR effect and tracking of nanoparticles [66]. The Brownian motion of the NPs in one dimension adds up over the RMHD trajectories and is perpendicular to the RMHD trajectory, producing a diffusion coefficient. As shown in Figure 26, the LSPR and diffusion coefficient provide two orthogonal characterization modes, with each mode depending on the material composition, shape, and size and the interactions of the particles with the surrounding medium. The 82 nm AgNP and 140 nm SiO_2_@Au core–shell nanoparticles were identified by RMHD and DFM, respectively. This method can effectively characterize the individual, and population of, NPs through LSPR and diffusion coefficient modes, with each mode dependent on composition, shape, size, and interaction with the surrounding medium.

Chen et al. designed a cube plasmonic probe of AuNPs grown on the surface of CuO cubes, which allowed for quantitative detection of glucose under DFM monitoring, as shown in Figure 27 [67]. The average edge length of the CuO cubes was 50 nm, with a lattice spacing of 0.246 nm, and the diameter of the AuNPs was 5 nm. The plasmonic probe had good catalytic ability for glucose. The quantitative detection of glucose was achieved by detecting the plasmonic probe’s electrical signal, with a dynamic range of glucose detection between 0.16 mM and 5.6 mM, and an LOD of 4 μM. Zou et al. designed a Cu_2−x_Se nanoparticle (NP) plasmonic probe for detecting heparin (Hep) under DFM monitoring [10]. In the non-aggregated state, the LSPR scattering intensity of the Cu_2−x_Se NP plasmonic probe was very weak. The presence of Hep caused the aggregation of Cu_2−x_Se NP plasmonic probes encapsulated by CTAB. The aggregated Cu_2−x_Se NP plasmonic probes showed a significantly increased LSPR scattering intensity around 510 nm, and the intensity was linearly correlated with the concentration of Hep, as shown in Figure 27b. The LOD of the plasmonic probe for Hep detection was 4 ng/mL.

Lin et al. synthesized Janus-structured nanocomposites by connecting 3-mercaptopropionic acid (3 MPA) with 4 nm ZnO quantum dots (ZnO QDs) and 6 nm AuNPs [68]. Unmodified ZnO QDs showed weak LSPR scattering in the UV range at 380 nm and strong LSPR scattering in the visible range at 530 nm. The LSPR scattering of ZnO QD/AuNP almost disappeared in the visible range, while it sharply increased in the UV range at 380 nm. This was due to the defect emission of the ZnO QDs being available to excite the LSPR of AuNPs. The excited electrons in the AuNPs remained in a higher energy state, leading to accumulated electrons being transferred to the conduction band of ZnO QDs. Therefore, an additional radiative bandgap transition was obtained when the excited electrons recombined with the holes in the valence band. This result provided direct evidence for the LSPR-enhanced UV emission mechanism of ZnO QDs, which can be useful in designing high-efficiency metal-semiconductor nanocomposites for photonics applications. Table 2 shows the recent advances in the application of multicomponent composite plasmonic nanomaterials.

## 4. Conclusions

The application status of plasmonic nanoparticles in dark-field sensor systems is examined in this work, covering LSPR theory, detection principles, properties of nanomaterials with various morphologies and structures, and related applications. In conclusion, the extremely appealing features of plasmonic nanomaterials in dark-field sensor systems suggest intriguing applications for plasmonic nanosensor platforms. The advancement of dark-field microscopy imaging technologies is inextricably linked to the advancement of novel materials and production techniques. In addition to single-component nanomaterials, the present work provides concepts and methods for increasing the application of plasmonic nanoparticles and dark-field sensors by mixing at least two materials and using their synergistic impact. Furthermore, new functional plasmonic nanomaterials will be introduced in the future, such as aluminum plasmon, plasmonic molecules, and plasmonic chiral materials. We believe that future plasmonic nanomaterials research should concentrate on the following aspects:

(1) From a fundamental standpoint, developing analytical theoretical models capable of rigorously describing plasmonic coupling in plasmonic nanomaterial components is critical for their utilization in a wide range of applications. In comparison to sophisticated computational methods, a theoretical model can give the physical basis for numerous plasmonic coupling phenomena, as well as the foundation for the discovery of new plasmonic-related phenomena and applications.

(2) One of the next research goals will be to create effective ways for controlling the synthesis and assembly of plasmonic nanomaterials, the majority of which are now generated using wet chemical, seed-mediated procedures. Many optoelectronic devices require nanoparticles in various sequences and combinations. Only time-consuming and expensive top-down techniques can currently generate nanoarrays with regulated nanoparticle geometry and orientation. Photolithography nanoparticles, on the other hand, are often composed of tiny gold nanoparticles. This polycrystalline nature causes significant plasmonic damping, which reduces plasmonic resonance. Wet chemistry nanoparticles, on the other hand, are mainly single crystals. As a result, simple and highly controlled approaches are critical for the future.

It may be projected that the development of new materials and technologies will encourage the continual improvement and development of dark-field microscopy systems, and that the field of application of dark-field microscopy systems will become increasingly broad in the future. Moreover, as this intriguing and promising sector develops, numerous exciting scientific discoveries and technology applications are certain to emerge.

## Figures and Tables

**Figure 1 nanomaterials-13-02027-f001:**
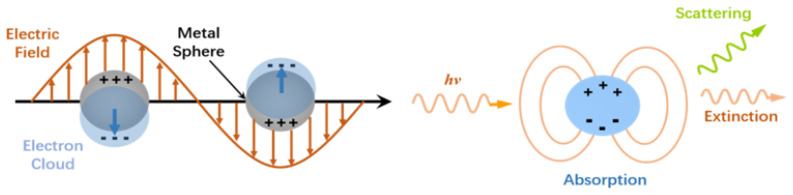
Schematic diagram of LSPR.

**Figure 2 nanomaterials-13-02027-f002:**
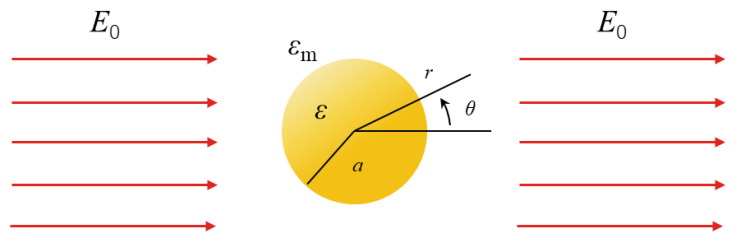
Schematic diagram of metal nanoparticles in an electrostatic field.

**Figure 3 nanomaterials-13-02027-f003:**
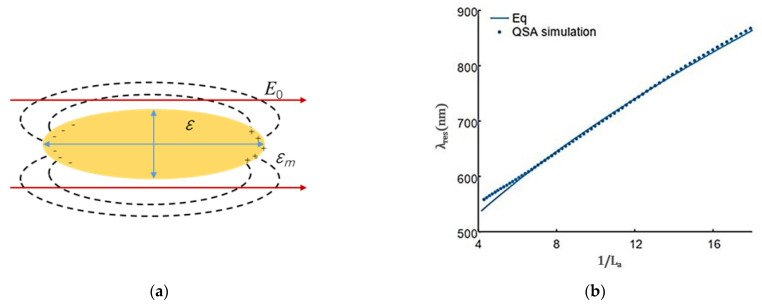
Quasi-static approximation of ellipsoidal metal nanoparticles. (**a**) Schematic diagram of a metal ellipsoid particle in a uniform electrostatic field, and (**b**) resonance wavelength of a gold ellipsoid particle calculated by the Drude model and quasi-static approximation.

**Figure 4 nanomaterials-13-02027-f004:**
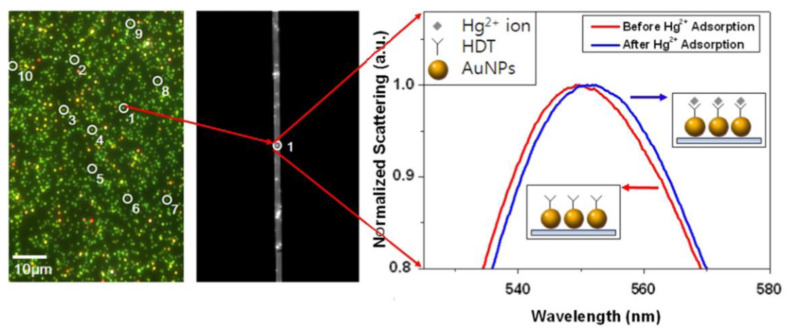
Application examples of the mechanical scanning dark-field micro-imaging system. Reprinted with permission from Ref. [18].Copyright 2010 IOP Publishing.

**Figure 5 nanomaterials-13-02027-f005:**
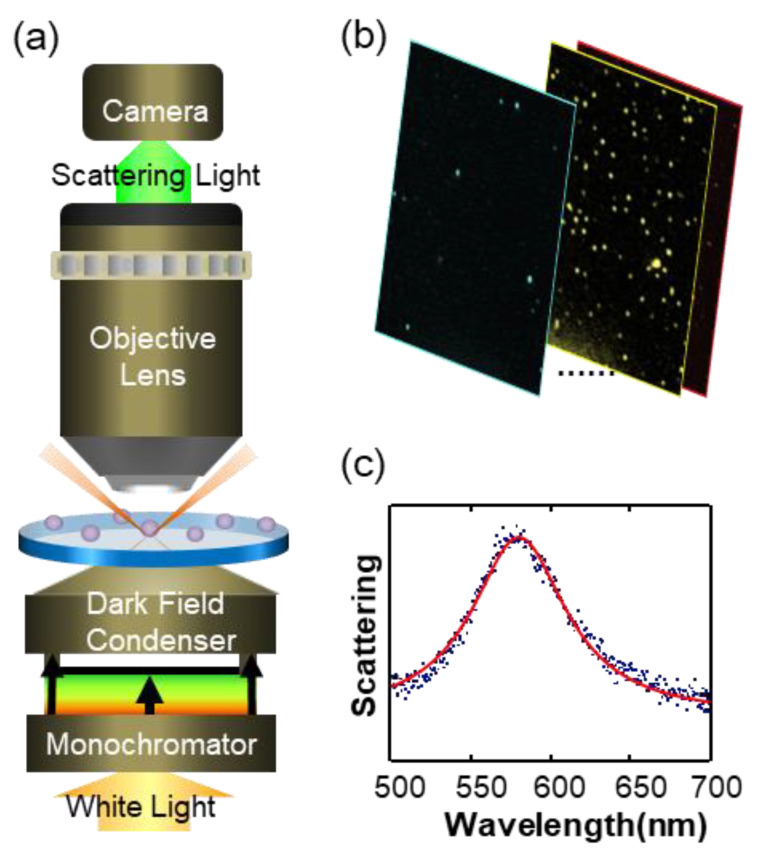
Dark-field micro-imaging system with incident wavelength modulation. (**a**) System structure, (**b**) image data set, and (**c**) The blue dots are the measured values and the red lines are spectral fits. Reprinted with permission from Ref. [19]. Copyright 2020 Elsevier.

**Figure 6 nanomaterials-13-02027-f006:**
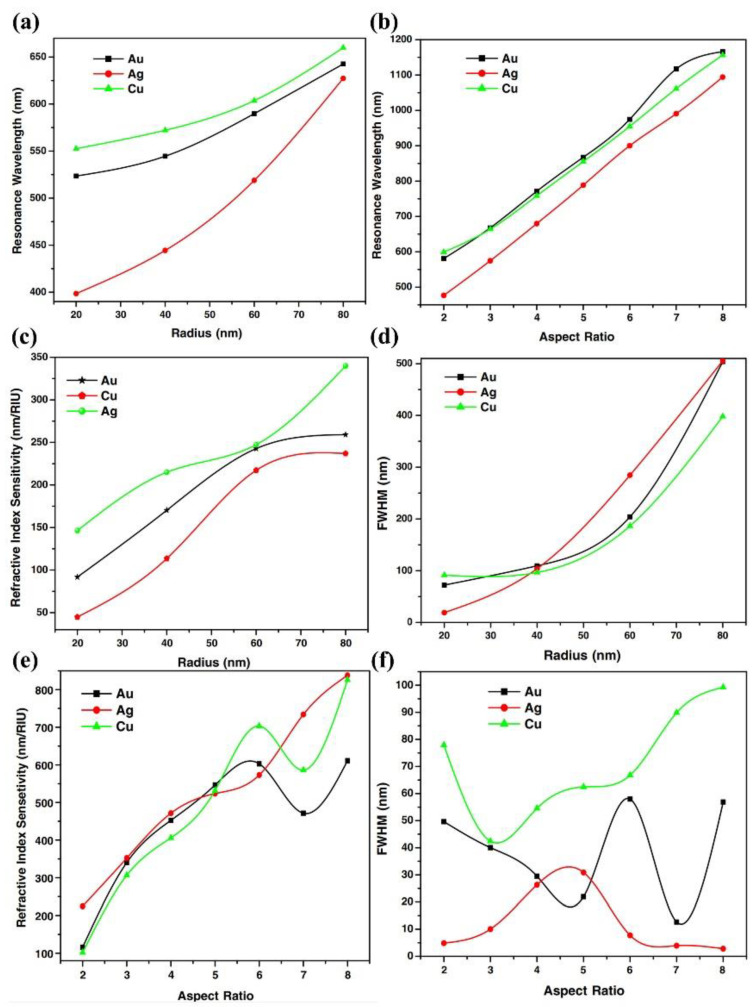
The relation of plasmonic resonance wavelength to diameter and aspect ratio [20]. (**a**) Resonance wavelength to the diameter of Au, Ag, and Cu spherical nanoparticles. (**b**) Resonance wavelength to the aspect ratio of Au, Ag, and Cu rod-shaped nanoparticles. (**c**) Refractive index sensitivity to the diameter of Au, Ag, and Cu spherical nanoparticles. (**d**) FWHM to the diameter of Au, Ag, and Cu spherical nanoparticles. (**e**) Refractive index sensitivity to the aspect ratio of Au, Ag, and Cu rod-shaped nanoparticles. (**f**) FWHM to the aspect ratio of Au, Ag, and Cu rod-shaped nanoparticles. Reprinted with permission from Ref. [20]. Copyright 2011 MDPI AG.

**Figure 7 nanomaterials-13-02027-f007:**
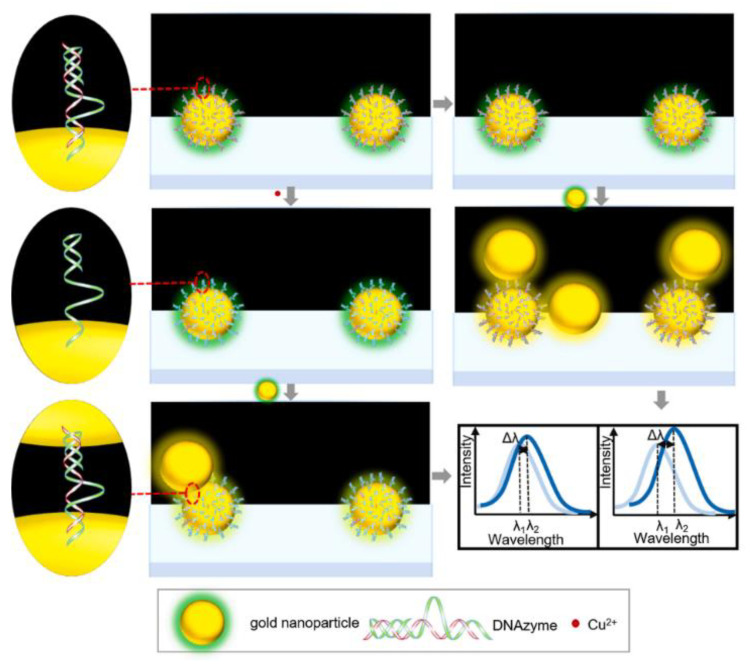
Schematic illustration of the procedure for detection of copper ion using DNAzyme-modified AuNPs. The gray arrows in the middle column connect the experiment stages in the case of high Cu^2+^ concentration, while the right side displays relatively low or no concentration. Reprinted with permission from Ref. [21]. Copyright 2021 Elsevier.

**Figure 8 nanomaterials-13-02027-f008:**
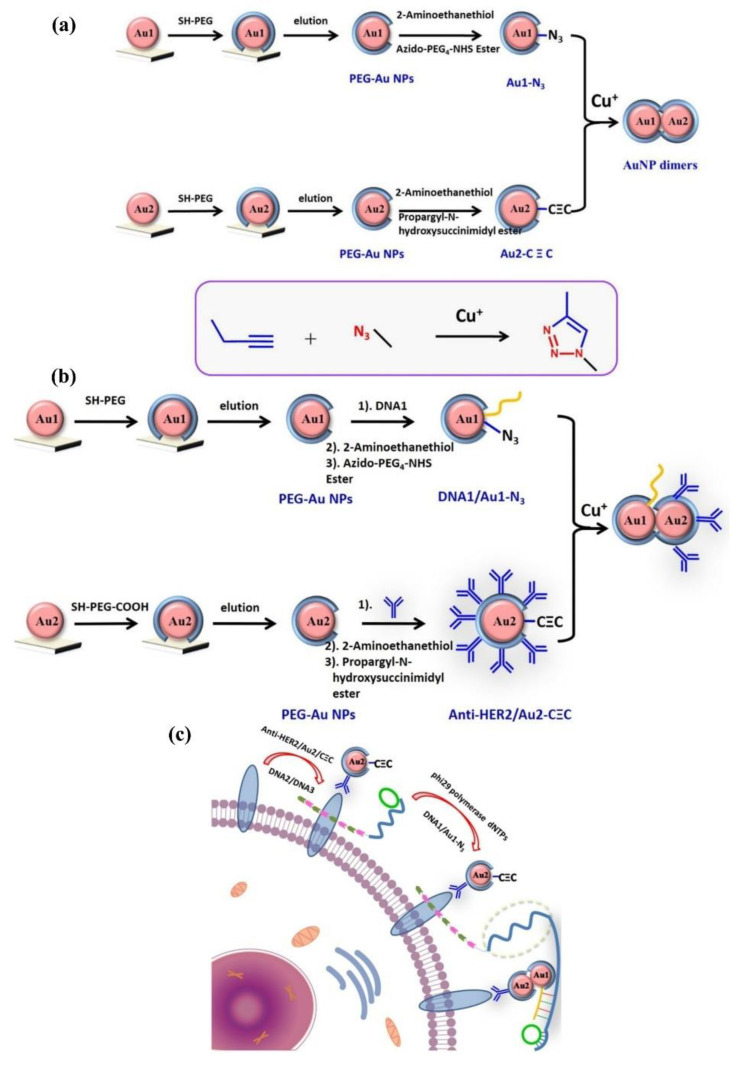
Schematic diagram of three schemes for Cu^2+^ detection with AuNPs. Reprinted with permission from Ref. [23]. Copyright ©2020 American Chemical Society. (**a**) Schematic diagram of single Cu^+^ to form AuNP dimers. (**b**) Schematic representation of Cu^+^-catalyzed click chemistry to form AuNP dimers. (**c**) Schematic representation of AuNP nanoaggregates for the HER2 protein.

**Figure 9 nanomaterials-13-02027-f009:**
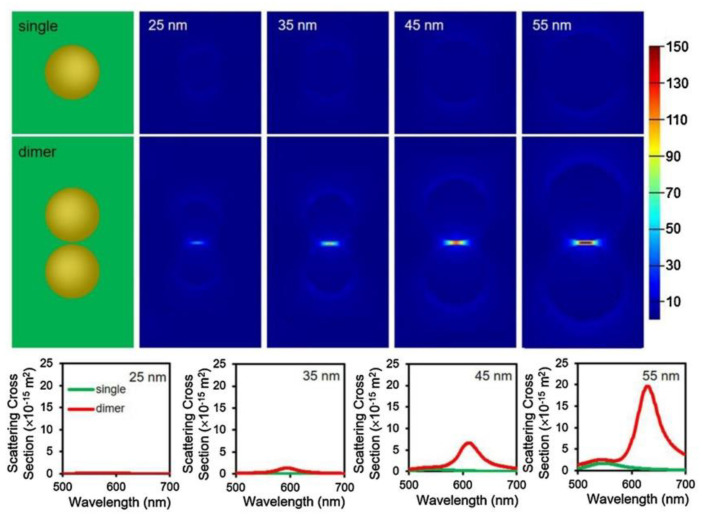
Simulation of the electrical field distribution from single GNPs of different sizes (25 nm, 35 nm, 45 nm and 55 nm) and GNP dimers. Reprinted with permission from Ref. [24]. Copyright 2020 Elsevier.

**Figure 10 nanomaterials-13-02027-f010:**
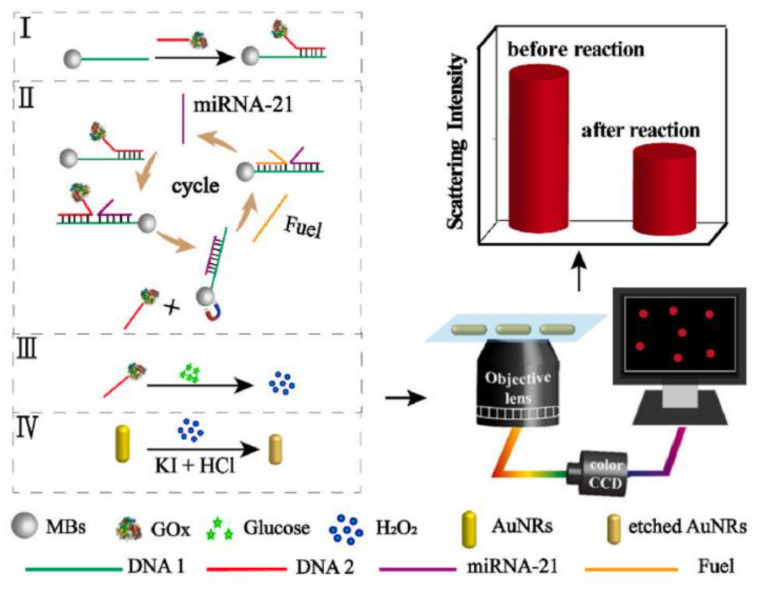
Schematic diagram of high sensitivity detection of miRNA-21 with the strand displacement amplification strategy under DFM. I–IV in the figure represent the four steps of the the strategy for detection of miRNA. Reprinted with permission from Ref. [31]. Copyright 2022 Elsevier.

**Figure 11 nanomaterials-13-02027-f011:**
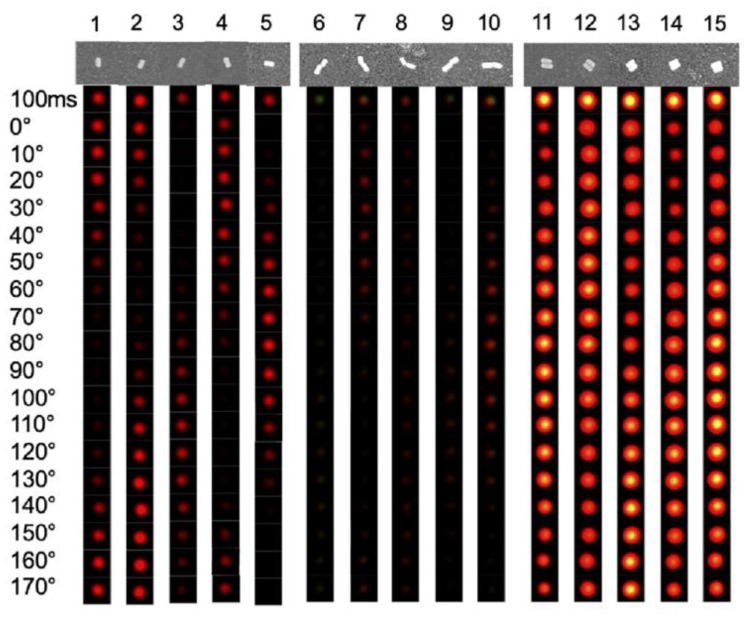
SEM and DFM images of AuNRs in three states. (Above) SEM images of single AuNRs (particles 1–5), end-to-end dimers (particles 5–10), and parallel dimers (particles 11–15). (Below) Corresponding DFM images of AuNRs in three states at different angles under polarization. Reprinted with permission from Ref. [33]. Copyright ©2023 American Chemical Society.

**Figure 12 nanomaterials-13-02027-f012:**
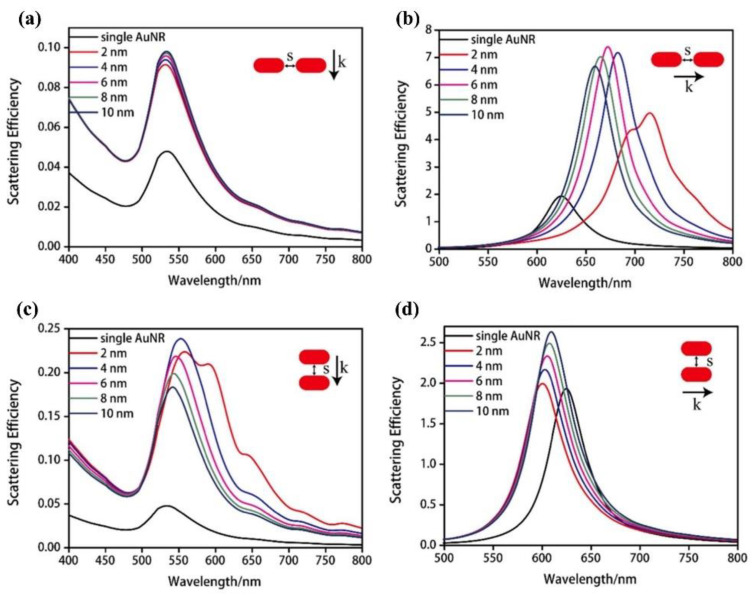
Simulated scattering spectrum of the end-to-end AuNR dimer of the (**a**) parallel and (**b**) longitudinal plasmon resonance mode with various inter-nanorod distances. Simulated scattering spectrum of the side-by-side AuNR dimer of the (**c**) parallel and (**d**) longitudinal plasmon resonance mode with various inter-nanorod distances. Scattering efficiency in this figure is defined as the ratio of scattering cross-section to the geometry cross-section. “k” in these figures refers to the direction of the wave vector and “s” refers to the spatial distance between two AuNRs. Reprinted with permission from Ref. [34]. Copyright 2018 Elsevier.

**Figure 13 nanomaterials-13-02027-f013:**
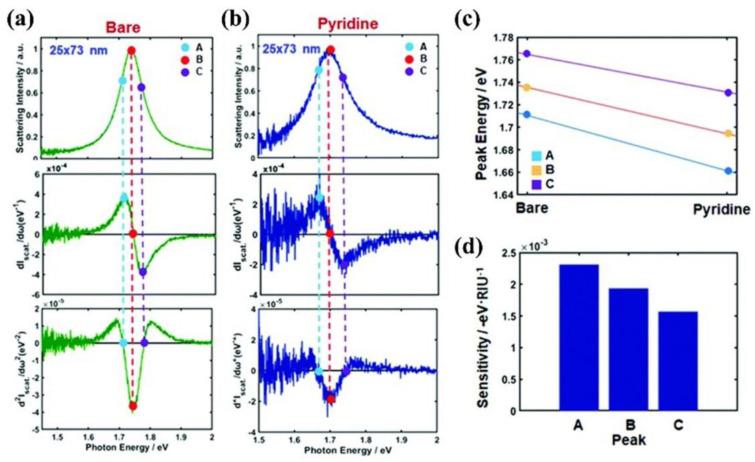
Inflection point method for single-particle LSPR scattering sensing with AuNRs in the presence of pyridine in water. (**a**,**b**) LSPR scattering efficiencies (first row) and its first− (second row) and second− (third row) order derivatives. (**c**) Peak energy plotted against the chemical adsorption of pyridine for points A, B, and C. (**d**) Detection sensitivity on the peak shifts of A, B, and C. Reprinted with permission from Ref. [35]. Copyright 2021, Royal Society of Chemistry.

**Figure 14 nanomaterials-13-02027-f014:**
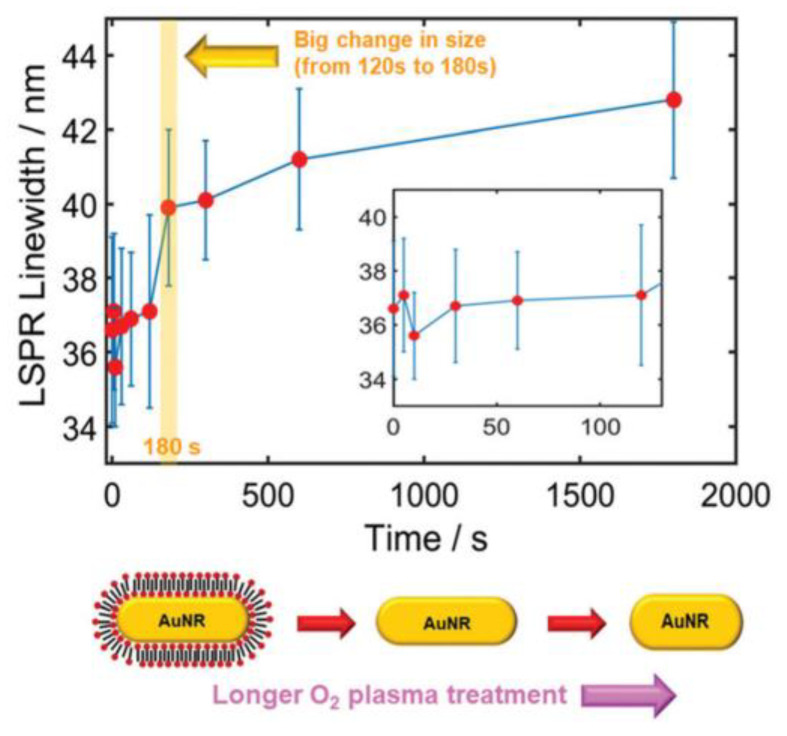
LSPR linewidth of AuNRs versus plasmonic treatment time [36]. The inset is an enlargement of the main plot in the range of 0–120 s. Copyright 2020, Royal Society of Chemistry.

**Figure 15 nanomaterials-13-02027-f015:**
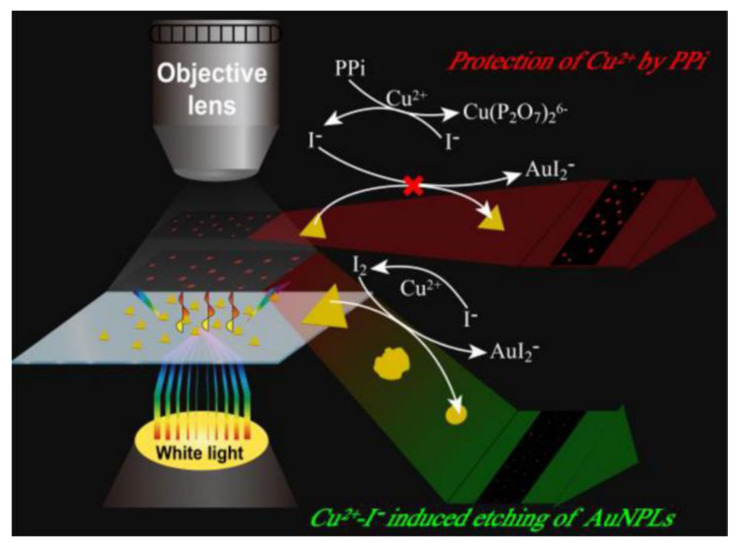
Working principle of AuNPLs for the PPi assay. Reprinted with permission from Ref. [37]. Copyright ©2020 American Chemical Society.

**Figure 16 nanomaterials-13-02027-f016:**
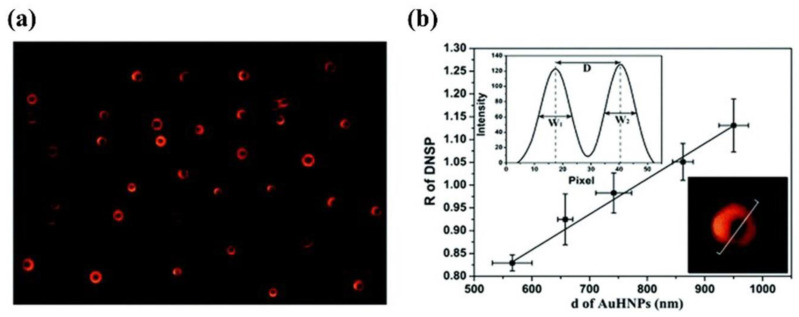
The DNSP properties of AuHNPs. Reprinted with permission from Ref. [38]. Copyright 2018, Royal Society of Chemistry. (**a**) DFM image (DNSPs) of different sizes of AuHNPs larger than 500 nm. (**b**) Plot depicting the linear relationship between the degree of peak separation (R) and the size of AuHNPs (d, the distance between the two opposite edges). Error bars are calculated from the measurements of five nanoplates; r^2^ = 0.991.

**Figure 17 nanomaterials-13-02027-f017:**
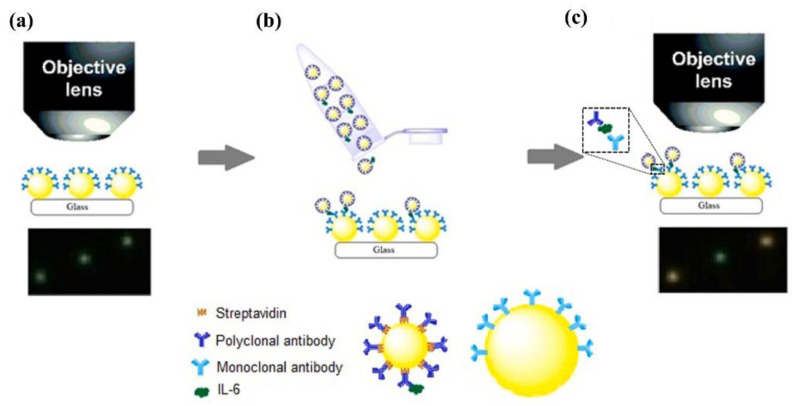
Schematic workflow of the experiment. Reprinted with permission from Ref. [43]. (**a**) Cover slip surface with immobilized 67 nm AuNPs modified with monoclonal antibody under DFM are seen as green dots. (**b**) Addition of reporter nanoparticles containing the antigen to the surface. Reporter nanoparticles containing antigen will form core–satellite assemblies with the 67 nm AuNPs immobilized on the surface. (**c**) The coverslip surface under DFM after addition of reporter nanoparticles and formation of core–satellite assemblies. The formation of assemblies will be evident by a shift in the hue of the nanoparticles to an orange/reddish color. Copyright 2020 Elsevier.

**Figure 18 nanomaterials-13-02027-f018:**
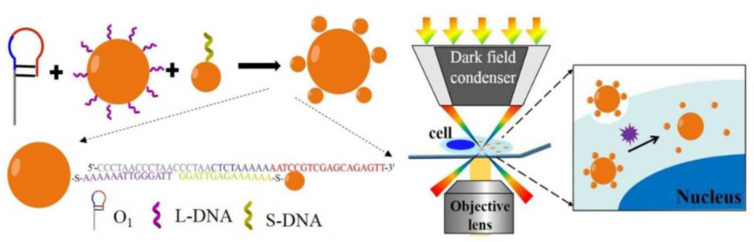
Schematic diagram of in situ analysis of intracellular telomerase activity using AuNPs. Reprinted with permission from Ref. [41]. Copyright ©2017 American Chemical Society.

**Figure 19 nanomaterials-13-02027-f019:**
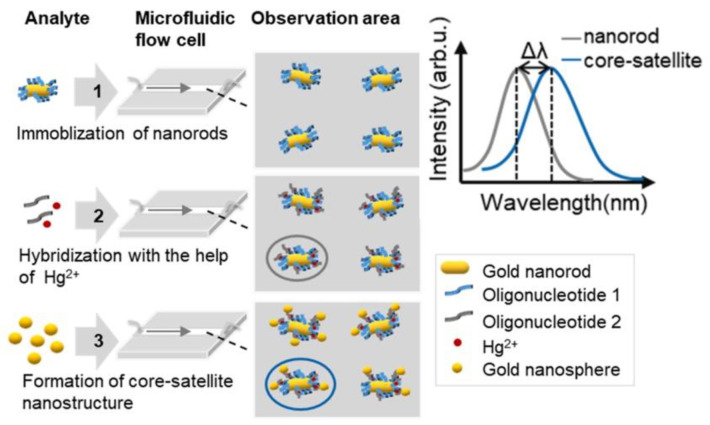
Steps to in situ detection of Hg^2+^ using the core–satellite nanostructure under a DFM. Reprinted with permission from Ref. [45]. Copyright 2023 Elsevier.

**Figure 20 nanomaterials-13-02027-f020:**
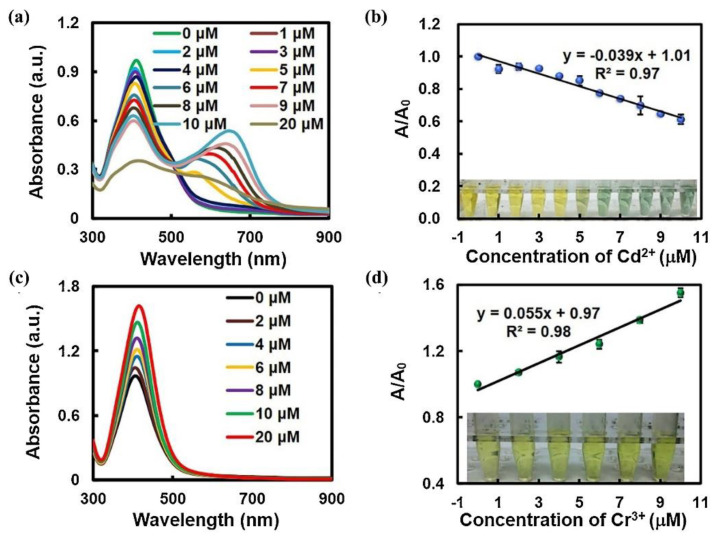
(**a**) UV−vis absorption spectra of Au@Ag NPs with Cd^2+^ (from 0 to 20 μM). Reprinted with permission from Ref. [54]. (**b**) The linear responses to Cd^2+^ ranging from 1 to 10 μM. A0 and A are the intensities of absorbance before and after adding Cd^2+^, respectively. Inset: the corresponding colorimetric changes of Au@Ag NP solutions under different Cd^2+^ concentrations. (**c**) UV−vis absorption spectra of Au@Ag NP solutions with Cr^3+^ (from 0 to 20 μM). (**d**) The linear responses to Cr^3+^ ranging from 2 to 10 μM. A0 and A are the intensities of absorbance before and after adding Cr^3+^, respectively. Inset: the corresponding colorimetric changes of GNP@Ag NP solutions under different Cr^3+^ concentrations. Copyright 2021 Elsevier.

**Figure 21 nanomaterials-13-02027-f021:**
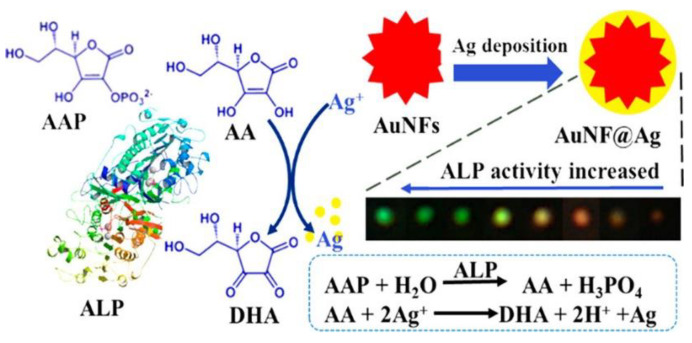
Diagram of AuNFs@Ag detection of AAP. Reprinted with permission from Ref. [56]. Copyright ©2018 American Chemical Society.

**Figure 22 nanomaterials-13-02027-f022:**
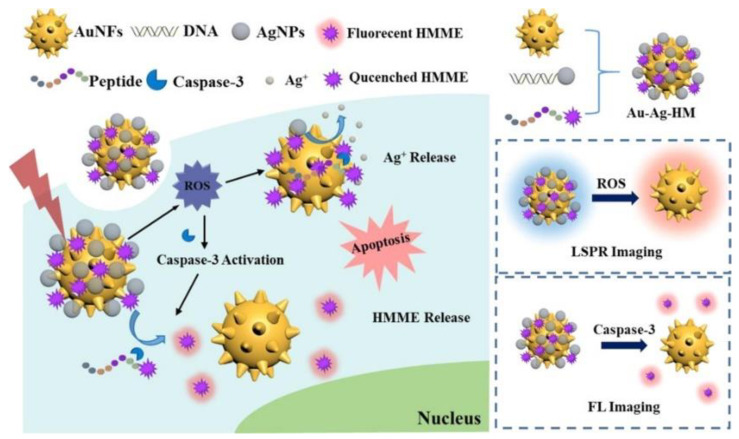
Diagram of AuNFs@Ag detection of AAP. Reprinted with permission from Ref. [57]. Copyright ©2018 American Chemical Society.

**Figure 23 nanomaterials-13-02027-f023:**
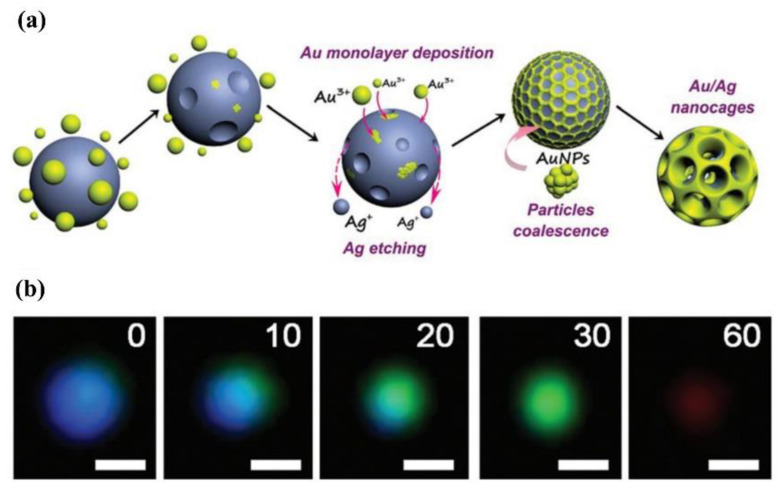
Schematic illustration of the real-time monitoring of the GE reaction using a light scattering DFM imaging system. Reprinted with permission from Ref. [58]. Copyright 2018, Royal Society of Chemistry. (**a**) The mechanism of GE involved a reaction between individual AgNPs and Au^3+^ in the HAuCl4 solution (the blue particles represent AgNPs, while the little yellow particles correspond to Au^3+^ ions). (**b**) DFM images of the GE process at the stage represented in (**a**). The times to obtain DFM images during the GE process are 0, 10, 20, 30, and 60 min, respectively. The scale bar is 100 nm for all images.

**Figure 24 nanomaterials-13-02027-f024:**
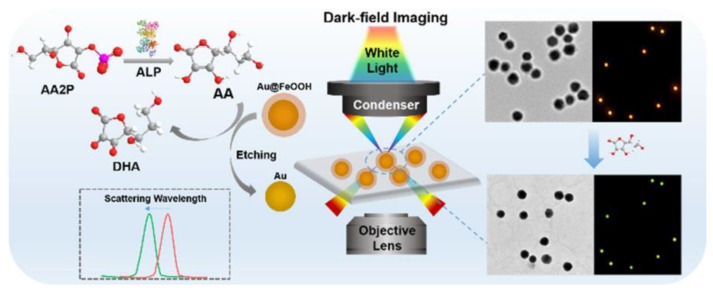
Diagram of Au@FeOOH detection of ALP. Reprinted with permission from Ref. [63]. Copyright ©2021 American Chemical Society.

**Figure 25 nanomaterials-13-02027-f025:**
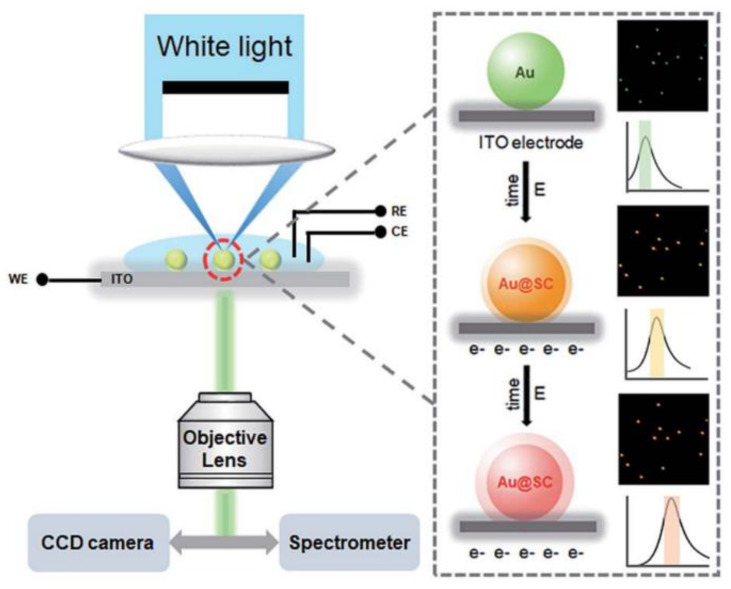
Experimental setup of the real-time DFM imaging of in situ electrochemical synthesis of Au@SC core–shell NPs. The color bar graph on the right corresponds to the color of the scattered light of the nanoparticles under a dark-field microscope. Reprinted with permission from Ref. [64]. Copyright 2019, Royal Society of Chemistry.

**Figure 26 nanomaterials-13-02027-f026:**
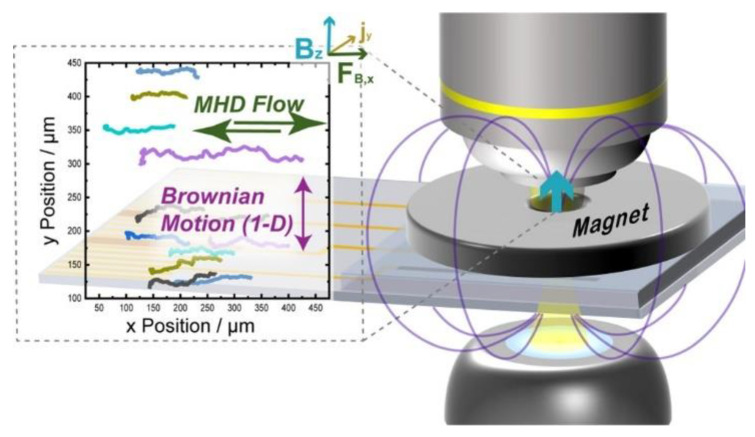
Characterization of nanoparticles in diverse mixtures by DFM with redox magnetohydrodynamics microfluidics. Reprinted with permission from Ref. [66]. Copyright © 2022 The Authors. Published by the American Chemical Society.

**Figure 27 nanomaterials-13-02027-f027:**
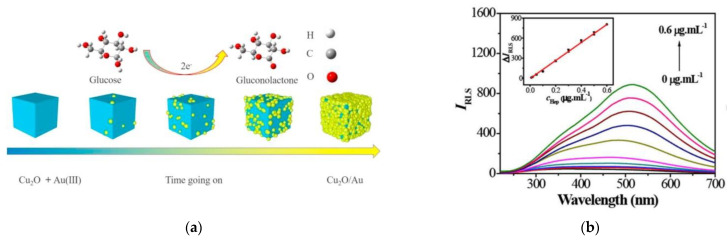
(**a**) Schematic illustration of Cu2O/Au NPs for non−enzymatic glucose detection. Reprinted with permission from Ref. [67]. Copyright 2021 Elsevier. (**b**) The sensitivity of Cu_2−x_Se for Hep. Reprinted with permission from Ref. [10]. Copyright 2020 Elsevier.

**Table 1 nanomaterials-13-02027-t001:** Application of single-component plasmonic materials.

Structure	Nanoparticle	Probe	Analyte	Detection Range	LOD	Ref.
nanospheres	AuNPs	DNA	Cu^2+^	0.1 Nm−5 μM	0.082 nM	[21]
	AuNPs	MPBA	galactose	1−75 nM	0.83 nM	[24]
	AuNPs	Tetrazine, trans−cycloctene	ATP	-	-	[46]
	AuNPs	oligonucleotide	Hg^2+^	0.005−25.0 nM	1.4 pM	[22]
	AuNPs	dsDNA	PARP−1	0.2−10 mU	-	[44]
	AuNPs	DNA	miRNA−122	100 pm−100 nM	-	[47]
nanorods	AuNRs	CTAB	telomerase	100−24,000 cells	43 cells	[48]
	AuNRs	GADD45	p53 protein	10−10^6^ fM	11.47 fM	[29]
	AuNRs	antibody	TNF−α, IL−4, IL−6, IL−10	-	-	[30]
	AuNRs	DNA	miRNA−21	50−2500 fM	1.72 fM	[33]
	AuNRs	DNA	miRNA−Let−7a	2−2000 fM	0.53 fM	[33]
	AuNRs	DNA	DNA in serum	0.1 pM−1 nM	30 fM	[34]
	AuNRs	OTA Aptamer, Poly A	OTA	0.1 nM−10 μM	<1 nM	[49]
	AuNRs	-	miRNA−21	0.1 pM−10 nM	71.22 fM	[31]
	AuNRs	PDMA, PCM−b−PEG	Bacterial lipase	-	-	[50]
nanosheets	AuNTs	-	PPi	7–100 nM	1.09 nM	[37]
	AuNTs	oligonucleotides	DNA	-	-	[51]
	AuHNPs	-	Au@Hg	-	-	[38]
	AuBPs	p−ATP@Biotin	Streptavidin	-	-	[52]
	AuBPs	p−ATP@Anti−IgG	IgG	-	-	[52]
Nanocore–satellite	AuNPs	L−DNA, S−DNA	telomerase	3.8 × 10^−13^−1.9 × 10^−11^ IU	1.3 × 10^−13^ IU	[41]
	AuNPs	Anti IL−6	IL−6	-	0.01 ng/mL	[43,53]
	AuNR, AuNPs	oligonucleotides	Hg^2+^	10 pM−10 μM	2.7 pM	[45]

**Table 2 nanomaterials-13-02027-t002:** Application of composite plasmonic materials.

Structure	Nanoparticle	Probe	Analyte	Detection Range	LOD	Ref.
composite plasmonic materials	AuNP@Ag	-	Cd^2+^	1−5 μM	11.5 nM	[54]
	AuNP@Ag	-	Cr^3+^	2−10 μM	26.8 nM	[54]
	AuNFs, Ag^+^	-	ALP	0.1−60 μU L^−1^	0.03 μU L^−1^	[56]
	Au−Ag−HM	-	ROS	-	-	[57]
	Au−Ag−HM	-	caspase−3	0.05−20 nM	26.7 pM	[57]
	AgNPs, Au^3+^	-	GE	-	-	[58]
	AuNPs@MnO_2_	-	glucose	0.05−20 mM	12.9 nM	[61]
	AuNP@FeOOH	-	ALP	0.2−6.0 U/L	0.06 U/L	[63]
	AgSHINs	-	Hg^2+^	10^−10^−10^−4^ M	-	[65]
	AuNR@Ag	GOx protein	glucose	5−100 μM	0.5 μM	[69]
	Au@Ag NCs	D-mannose	ConA	10 nM−10 μM	2 nM	[70]
	AuNSs, Ag^+^	-	MAO-B	0.05−1 μg mL^−1^	8.0 ng mL^−1^	[71]
	Au@Ag CSN	-	PtCl_6_^2−^	-	-	[72]
	AuNP@MnO_2_	-	ALP	0.06−0.48 mU/mL	5.8 μU/mL	[60]
	Au/Ag NCs	oligonucleotides	miRNA−21	-	-	[59]
	Au@AgI	-	S^2+^	0.1−500 nM	33 pM	[62]
	Cu_2_O/Au	-	glucose	0.16−5.6 mM	4 mM	[67]
	AuNP@Ag	-	MnO_4_^−^	0−6 μM	46 nM	[73]
	Cu_2−x_Se NPs	CTAB	Hep	0.01−0.6 μg mL^−1^	4.0 ng mL^−1^	[10]
	ZnO QD/AuNP	-	-	-	-	[68]

## Data Availability

Not applicable.

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
