# Peer review of "Plasmonic Nanomaterials in Dark Field Sensing Systems"

_nanomaterials, 2023, doi:10.3390/nano13132027_

Round 1

Reviewer 1 Report

This review describes research on biosensing by dark-field microspectroscopy using localized surface plasmons generated in nanostructures such as noble metals. While it provides systematic coverage and useful information for the reader with respect to the applications presented, the following points seem to require revision.

1. First, the use of the term "plasma nanomaterials" seems possibly misleading. Presumably, the authors intend to refer to “plasmonic nanomaterials”, but the term "plasma nanomaterials" could also refer to nanomaterials fabricated using plasma-related technologies. Since plasma and plasmon seem to refer to slightly different phenomena, perhaps phrases like "plasma nanomaterials" in the title and text should be replaced with something like "plasmonic nanomaterials"?

2. The contents of this review seem to focus mainly on biosensing applications using DFM. In Section 2, there is an explanation of the principles of LSPR and DFM, but the DFM part seems to be more focused on the practical aspects of the subject. I think it would be better to have a little more explanation of the conceptual part of biosensing using LSPR before the part on DFM. There should be a clearer explanation regarding the benefits of using DFM as opposed to biosensing without DFM.

Author Response

Comment1: First, the use of the term "plasma nanomaterials" seems possibly misleading. Presumably, the authors intend to refer to “plasmonic nanomaterials”, but the term "plasma nanomaterials" could also refer to nanomaterials fabricated using plasma-related technologies. Since plasma and plasmon seem to refer to slightly different phenomena, perhaps phrases like "plasma nanomaterials" in the title and text should be replaced with something like "plasmonic nanomaterials"?

Response1: We appreciate the reviewer’s comments. Considering the reviewer’s suggestion, "plasma" in the title has been revised to "plasmonic". We have also made corresponding changes in the text.

Comment2: The contents of this review seem to focus mainly on biosensing applications using DFM. In Section 2, there is an explanation of the principles of LSPR and DFM, but the DFM part seems to be more focused on the practical aspects of the subject. I think it would be better to have a little more explanation of the conceptual part of biosensing using LSPR before the part on DFM. There should be a clearer explanation regarding the benefits of using DFM as opposed to biosensing without DFM.

Response2: Thank you for your valuable comments. The biosensor consists of the molecular recognition part (sensitive part) and the signal converter (transducer). The molecular recognition part is usually the specific ligand probe of the molecule to be measured, and the transducer is the nanoparticle. As a detecting instrument, DFM can obtain the photoelectric signal of nanoparticles. In fact, utilizing standard spectrometers to obtain the spectrum of nanoparticles is relatively convenient. The advantage of utilizing DFM is that it allows us to see how the spectrum of a single nanoparticle changes. Clearly, the application of DFM can improve sensitivity and LOD, as well as analyze the chemical binding process by changing the spectrum of single-particle nanoparticles. For details, see the first paragraph of Section 2.2 we added. To avoid verboseness, part of the second paragraph of Part 3 has been deleted.

Reviewer 2 Report

Authors reviewed the local surface plasmon resonance (LSPR) excitation mechanism of plasma nanoprobes and its critical significance in the control of dark field sensing, as well as main sensing strategies based on plasma nanomaterial dielectric environment modification, electromagnetic coupling, and charge transfer. However this manuscript is not sufficient to be published in Nanomaterials.

Authors should revise the following:

1) Merits of dark field sensing compared to conventional Raman spectroscopy or surface-enhanced Raman spectroscopy. In conventional Raman spectroscopy or surface-enhanced Raman spectroscopy, excitation light can be blocked by notch filter or monochromator.

2) Plasma nanomaterials should be replaced with other words. Generally, plasma represents the device to produce reactive oxygen and nitrogen species.

3) The meaning of manuscript title is unclear. The manuscript title should be changed to the title that corresponds the manuscript contents. In manuscript, authors described the effects of nanoparticle shape on the optical sensing properties of nanomaterials.

4) Authors included the effects of aspect ratio and the particle size on the surface plasmon resonance wavelength. To provide meaningful review to readers, it should include the effects of aspect ratio and the particle size on the biosensing and optical imaging.

Extensive editing of English language required.

Author Response

Comment1: Merits of dark field sensing compared to conventional Raman spectroscopy or surface-enhanced Raman spectroscopy. In conventional Raman spectroscopy or surface-enhanced Raman spectroscopy, excitation light can be blocked by notch filter or monochromator.

Response1: According to the reviewer's comments, we have added the following content in part 2.2: “Plasmonic nanoparticles have been widely used in surface-enhanced Raman scat-tering (SERS) imaging and dark field localized plasmon resonance (LSPR) imaging due to their special nano optical properties. SERS has ultra-high sensitivity and can detect liquid samples with ultra-low concentrations. However, due to its lack of quantitative detection capability and the fact that only a few basic metal particles with roughened surfaces (such as gold, silver, copper, lithium, sodium, and so on) could create SERS, the application range of SERS imaging was limited. Although various quantitative SERS analysis meth-ods have been developed in the laboratory, such as adding internal standards of known concentrations to the determinand, these methods have achieved excellent linear curves in small ranges, but have also made detection more difficult. In addition to quantitative detection, DFM is better than SERS in exploring mechanisms during the reaction process.” For details, see paragraph 2 of Section 2.2.

Comment2: Plasma nanomaterials should be replaced with other words. Generally, plasma represents the device to produce reactive oxygen and nitrogen species.

Response2: As the reviewer suggested that "plasma" in the title has been revised to "plasmonic". We have also made corresponding changes in the text. Thank you again for your professional advice.

Comment3: The meaning of manuscript title is unclear. The manuscript title should be changed to the title that corresponds the manuscript contents. In manuscript, authors described the effects of nanoparticle shape on the optical sensing properties of nanomaterials.

Response3: This manuscript describes gold, silver and other plasmonic nanomaterials and their latest applications in biochemical sensor detection using dark field microscopic imaging systems. The principle of LSPR and the detection method of DFM was elaborated in part 2. In part 3, we describes the recent research and applications of different plasmonic nanomaterials, which are divided into single-component materials and multicomponent materials. Among them, the single-component material is described according to the shape of nanosphere, nanorod, nano sheet, etc., and each shape also involves different materials, commonly used are gold, silver, and copper. Multicomponent nanomaterials are divided into bi-metallic materials and other multicomponent nanomaterials, which include a variety of morphologies and materials. Finally, we apologize for not having thought of a more appropriate title at this time. The current title may not be ideal, but it contains our content and theme. Thank you again for the reviewer's comments.

Comment4: Authors included the effects of aspect ratio and the particle size on the surface plasmon resonance wavelength. To provide meaningful review to readers, it should include the effects of aspect ratio and the particle size on the biosensing and optical imaging

Response4: Thanks to the reviewers for their professional comments, which greatly improved the quality of the manuscript. We have added the following to the last paragraph of section 3.1.1. “For plasmonic nanospheres, single-particle optical observation under DFM can detect silver nanoparticles as small as 20 nm and gold nanoparticles as small as 30 nm in diameter. According to the simulation and experimental verification, larger particle sizes typically exhibit higher peak shifts and peak intensities. However, increasing the particle size may cause the line broadening of the plasmon resonance peak, leading to the limitation of spectral resolution for wavelength changes. In other words, the increase in spectral full width at half maximum(FWHM) has a negative impact on sensor sensitivity. Therefore, choosing an appropriate nanoparticle size can effectively improve the detection sensitivity under DFM. Current research shows that AuNPs or AgNPs with a diameter of 40-80 nm are the optimal choice for spherical metal plasmonic nanoparticles for biochemical sensors.

We have added the following to the last paragraph of section 3.1.2.“The resonance wavelength of the long axis of nanorods is connected to the nanorods' length to diameter ratio, and the higher the ratio of length to diameter, the larger the reso-nance wavelength. In nanorod applications, the length-diameter ratio and particle size of nanorods are often chosen based on the best performance region of DFM. In general, na-norods with a length-diameter ratio of 2.5-3 are more typically utilized, and high optical imaging effects and detection sensitivity may be attained at this time.

Round 2

Reviewer 1 Report

The authors well revised the manuscript based on the referee report, I appreciate the authors' effort to improve the manuscript. Now, I recommend the manuscript suitable for publishing in this journal in its present form.